# BIGGER ISN'T ALWAYS MEMORIZING: EARLY STOPPING OVERPARAMETERIZED DIFFUSION MODELS

## ABSTRACT

Diffusion probabilistic models have become a cornerstone of modern generative AI, yet the mechanisms underlying their generalization remain poorly understood. In fact, if these models were perfectly minimizing their training loss, they would just generate data belonging to their training set, i.e., memorize, as empirically found in the overparameterized regime. We revisit this view by showing that, in highly overparameterized diffusion models, generalization in natural data domains is progressively achieved during training *before* the onset of memorization. Our results, ranging from image to language diffusion models, systematically support the empirical law that memorization time is proportional to the dataset size. Generalization vs. memorization is then best understood as a competition between time scales. We show that this phenomenology is recovered in diffusion models learning a simple probabilistic context-free grammar with random rules, where generalization corresponds to the hierarchical acquisition of deeper grammar rules as training time grows, and the generalization cost of early stopping can be characterized. We summarize these results in a phase diagram. Overall, our results support that a principled early-stopping criterion – scaling with dataset size – can effectively optimize generalization while avoiding memorization, with direct implications for hyperparameter transfer and privacy-sensitive applications.

## 1 INTRODUCTION

Diffusion models (Sohl-Dickstein et al., 2015; Ho et al., 2020) have recently emerged as a transformative paradigm in generative AI, enabling the synthesis of high-quality data across a wide range of modalities – images, videos, text, and complex 3D structures such as molecular conformations and protein sequences. Their strength lies in their scalability in generating diverse, high-fidelity samples by reversing a progressive noise addition process, making them both versatile and robust across domains. At the heart of this process is the estimation of a *score function* (Song & Ermon, 2019; Song et al., 2020): a noise-dependent vector field that guides denoising by pointing in the direction of increasing data likelihood. Since this function is learned from the empirical training distribution, minimizing the training loss optimally leads the model to reproduce the training data itself – a phenomenon known as *memorization* (Carlini et al., 2023; Somepalli et al., 2022). This phenomenon is observed in practical settings and raises significant privacy and copyright concerns, as models trained on sensitive or proprietary data may inadvertently regenerate such content, exposing private information or violating intellectual property rights (Wu et al., 2022; Matsumoto et al., 2023; Hu & Pang, 2023). In contrast, *generalization* corresponds to the model producing novel samples that are consistent with, but not identical to, the training data, thereby approximating the broader target distribution.

Despite the empirical success of diffusion models, the mechanisms underlying their ability to generalize remain poorly understood. A prevailing view – rooted in classical learning theory – is that generalization depends on *underparameterization* (Yoon et al., 2023; Zhang et al., 2023; Kadkhodaie et al., 2023): only models that lack the capacity to memorize their training data are expected to generalize. In this work, we go beyond this view by demonstrating that even heavily overparameterized diffusion models exhibit generalization during training *before* they start memorizing the training data. We systematically investigate this phenomenon, showing that generalization and memorization are not mutually exclusive but unfold as distinct temporal phases of training. Our main contributions are as follows:

- We empirically demonstrate the transition from generalization to memorization during training in a range of overparametrized diffusion models – including Improved DDPM (Nichol & Dhariwal, 2021), Stable Diffusion (Rombach et al., 2022), MD4 (Shi et al., 2024), and D3PM (Austin et al., 2021) – on both images and text data. We measure memorization and generalization metrics and systematically vary the training set size, showing that generalization improves gradually, before the onset of memorization.
- In all settings, we find the empirical law that the onset of memorization requires a number of training steps that is proportional to the training set size. In the appendix, we provide a theoretical scaling argument for kernel methods – including kernels corresponding to infinite-width neural networks – showing that a generic empirical score at fixed, low diffusion noise is learned with a training time proportional to the training set size.
- We study a discrete diffusion model trained to learn a simple *probabilistic context-free grammar*, where the number of training steps or samples required to generalize is known to be polynomial in the sequence length (Favero et al., 2025). We show that for moderate training set sizes, the diffusion model only learns the lowest levels of the hierarchical grammar rules – corresponding to partial generalization – before starting to memorize. For larger training set sizes, the onset of memorization appears after perfect total generalization is achieved. These results lead to a phase diagram for memorization and generalization as a function of sample complexity and time.

On the theoretical level, these findings call for a revision of the view of generalization in diffusion models as being solely determined by model capacity, showing that generalization arises *dynamically during training* in overparameterized diffusion models. On the practical level, our results suggest that early stopping and dataset-size-aware training protocols may be optimal strategies for preserving generalization and avoiding memorization as the size of diffusion models is scaled up. In fact, meeting privacy and copyright requirements with principled procedures is of utmost importance for the deployment of generative AI, in contrast to heuristic procedures that lack quantitative grounding (Dockhorn et al., 2022; Vyas et al., 2023; Chen et al., 2024).

## 2 DIFFUSION MODELS AND THE SCORE FUNCTION

Denoising diffusion models are generative models that sample from a data distribution $q(x_0)$ by reversing a noise addition process (Sohl-Dickstein et al., 2015; Ho et al., 2020; Song & Ermon, 2019; Song et al., 2020). The *forward process* generates a sequence of increasingly noised data $\{x_t\}_{1 \leq t \leq T}$, with distribution $q(x_1, \ldots, x_T | x_0) = \prod_{t=1}^{T} q(x_t | x_{t-1})$, where $t$ indicate the time step in a sequence $[0, \ldots, T]$. At the final time $T$, $x_T$ corresponds to pure noise. The *backward process* reverts the forward one by gradually removing noise and is obtained by learning the backward transition kernels $p_\theta(x_{t-1} | x_t)$ using a neural network with parameters $\theta$. Learning these backward kernels is equivalent to learning the *score function*, which is proportional to the conditional expectation $\mathbb{E}_{q(x_0|x_t)}[x_0]$. To learn the score function, the training is performed by minimizing a variational bound on the negative log likelihood of the data:

$$\mathbb{E}_{q(x_0)}[-\log p_\theta(x_0)] \leq \mathbb{E}_{q(x_0)}\left[-\log p_\theta(x_T) - \sum_{t=1}^{T} \log \frac{p_\theta(x_{t-1}|x_t)}{q(x_t|x_{t-1})}\right] := \mathcal{L}. \tag{1}$$

The loss $\mathcal{L}$ to learn the score function requires an integral over the target data distribution $q(x_0)$. In practice, this integral is estimated with a Monte Carlo sampling from $P$ training examples $\{x_0^{(i)}\}_{i \in [P]}$, associated with the empirical distribution $\hat{q}(x_0) = P^{-1} \sum_{i=1}^{P} \delta(x_0 - x_0^{(i)})$, where $\delta$ are Dirac deltas. Therefore, perfectly minimizing the empirical loss corresponds to learning the empirical score function, which generates $\hat{q}(x_0)$. As a result, diffusion models would only generate data of the training set, corresponding to *memorization*. Their generalization abilities, therefore, derive from not perfectly minimizing the empirical loss.

**Diffusion processes** For continuous data, like images, the forward process corresponds to time-discretized Gaussian diffusion with $q(x_t | x_{t-1}) = \mathcal{N}(x_t; \sqrt{1 - \beta_t} x_{t-1}, \beta_t \mathbb{I})$, where $\mathcal{N}$ represents the normal distribution and the sequence $\{\beta_t\}_{1 \leq t \leq T}$ is the variance schedule. For discrete data, several noising processes have been considered (Hoogeboom et al., 2021; Austin et al., 2021). The most popular for text is *masked diffusion with an absorbing state*, which progressively randomly masks tokens in the forward process. Another common choice is uniform diffusion, where in the forward process, tokens can flip to any other symbol with some probability depending on the noise level.

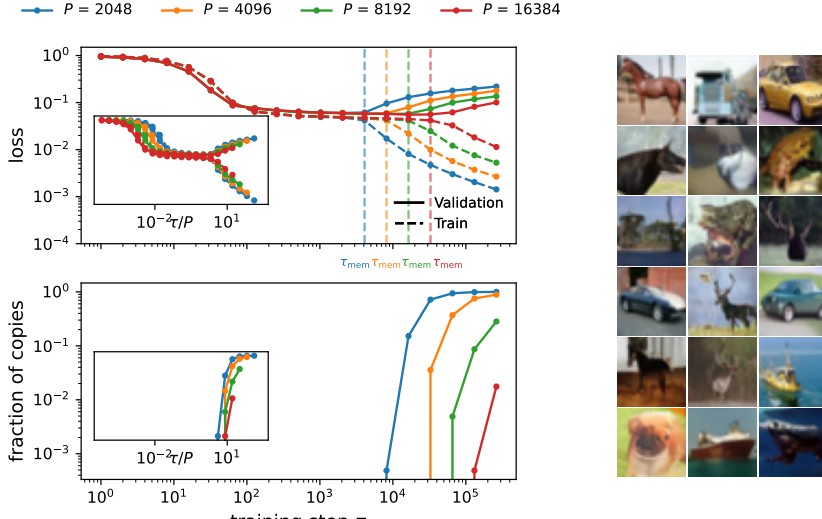

Figure 1: **Memorization dynamics in vision diffusion models.** *Left:* Train loss, validation loss, and fraction of copied images as a function of training steps $\tau$ for iDDPM models trained on CIFAR10 with varying training set sizes $P$. Both losses decrease initially, indicating generalization, but diverge at the onset of memorization ($\tau_{\mathrm{mem}}$), where the models start copying training data. Larger training sets delay $\tau_{\mathrm{mem}}$, scaling approximately linearly with $P$ (insets). *Right:* Samples generated with early stopping at $\tau_{\mathrm{mem}}$ with a model trained on $16,384$ images, achieving generalization and low FID. Further examples are presented in Appendix D

## 3 NUMERICAL EXPERIMENTS

### 3.1 VISION DIFFUSION MODELS

**Generalization before memorization**   We assess the generalization and memorization behaviors of vision diffusion models by considering Improved Denoising Diffusion Probabilistic Models (iD-DPMs) (Nichol & Dhariwal, 2021) with a U-Net architecture (Ronneberger et al., 2015; Salimans et al., 2017), including attention blocks (Vaswani et al., 2017). Each model, comprising approximately 0.5B parameters, is trained on four distinct subsets of the CIFAR-10 dataset (Krishnan et al., 2017), with training set sizes $P \in \{2048, 4096, 8192, 16384\}$. The models are trained for a total of 262,144 training steps, with full training details in Appendix B.

We track model performance using the diffusion losses on the train set and a validation set of $1,024$ images. At regular checkpoints, we generate $32,768$ images using each model, and evaluate memorization by calculating the fraction of generated images that are near-exact replicas of training samples. Specifically, following Carlini et al. (2023); Yoon et al. (2023), for a generated image $x$, we identify the two closest images $x'$ and $x''$ in Euclidean distance from the training set, and classify $x$ as a copy if $\|x - x'\|_2 / \|x - x''\|_2 < 1/3$. This threshold aligns with human perception of visual similarity (Yoon et al., 2023).

**Results and analysis**   Figure 1 (left panel) presents the results of this experiment. Our key findings are as follows:

1. **Generalization before memorization:** Initially, both train and validation loss decrease, indicating that the model is generalizing, i.e., approaching the population score. However, at some critical time $\tau_{\mathrm{mem}}$, the two losses bifurcate, signalling the onset of memorization. After this point, the number of copies among generated images steadily increases. By the end of training, all models exhibit some degree of memorization, with copy rates ranging from $1\%$ for the largest training set to $100\%$ for the smaller ones.

2. **Memorization is delayed by larger training sets:** The onset of memorization $\tau_{\mathrm{mem}}$ scales approximatively linearly with the training set size $P$, as indicated in the insets of Figure 1.

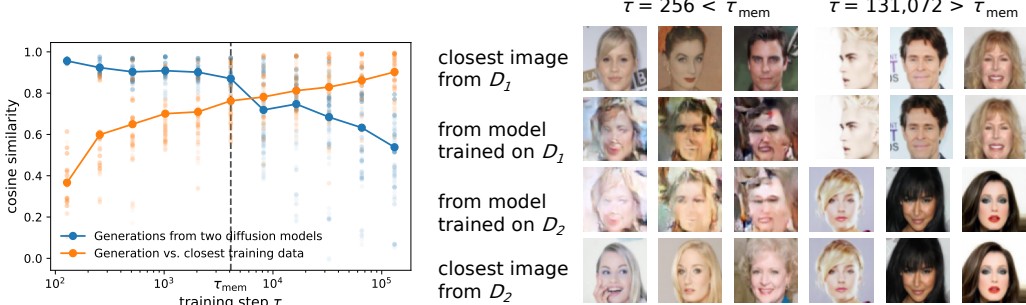

Figure 2: **Progressive generalization in vision diffusion models.** Cosine similarity between images generated by two diffusion models trained on disjoint subsets of CelebA of size $P = 2,048$, as a function of training steps $\tau$. Before memorization ($\tau < \tau_{\text{mem}}$), the models generate nearly identical images, indicating they are learning the same score function, and thus generalizing. After $\tau_{\text{mem}}$, the models diverge, generating images increasingly similar to their own training sets.

These observations suggest that early stopping can effectively prevent the model from entering the memorization phase. As a concrete example, the right panel of Figure 1 displays images generated by a diffusion model trained on 16,384 images, with early stopping applied. The quality and diversity of these images are quantified using the Fréchet Inception Distance (FID), calculated using Inception v3. The model achieves an FID score of $5.4$, indicating – despite being strongly overparameterized – robust generalization, while the rate of copies is $0\%$. In Appendix C, we show the same overfitting phenomenon in Stable Diffusion (Rombach et al., 2022) – a text-to-image latent diffusion model – fine-tuned on a subset of the LAION dataset (Schuhmann et al., 2022).

**Progressive generalization before memorization**   We extend our analysis by conducting a second experiment inspired by Kadkhodaie et al. (2023). Specifically, we train two models on two non-overlapping subsets $\mathcal{D}_1$ and $\mathcal{D}_2$ of $2,048$ images of CelebA (Liu et al., 2018), a dataset with faces of celebrities, each using an iDDPM (details in Appendix B). Our setup goes beyond prior work by dynamically tracking the evolution of the generated images throughout training, rather than statically only at convergence. This approach provides a detailed view of how models first approach the population score and then diverge after entering the memorization phase.

**Results and analysis**   We generate samples from both models at multiple checkpoints during training, initializing the generations from the same Gaussian random noise and fixing the stochastic part of the backward trajectories. Remarkably, initially, the images generated by the two models are nearly identical, reflecting that the two models are learning the same score function, even though they are trained on disjoint data subsets. However, at some time $\tau_{\text{mem}}$, the models begin to diverge. This divergence coincides with the onset of memorization, where the models start generating images increasingly similar to the ones contained in their respective training sets.

We quantitatively assess this phenomenon using cosine similarity between whitened images generated by the two models and their nearest training images. As shown in Figure 2:

1. **Before memorization** ($\tau < \tau_{\text{mem}}$), the two models generate nearly identical images, indicating that they are dynamically learning the same underlying distribution.

2. **During memorization** ($\tau > \tau_{\text{mem}}$), the similarity between the models' generated images decreases monotonically, while the similarity between each model's generated images and their own training set increases. This reflects the transition from generalization to memorization.

Our findings extend those of Kadkhodaie et al. by revealing that the transition from generalization to memorization is not only a matter of model capacity and final convergence but is dynamically observable throughout training. In practice, this further supports the view that early stopping can prevent the memorization phase and maintain generalization.

## 3.2 LANGUAGE DIFFUSION MODELS

We further extend our analysis of generalization and memorization to language data, using MD4, a masked diffusion model specifically designed for text (Shi et al., 2024). Our experiments are conducted on the text8 dataset, a standard benchmark for language modeling based on Wikipedia, with character-level tokenization. To the best of our knowledge, this is the first demonstration of memorization in the language diffusion setting. We train MD4 from scratch using a standard GPT-like transformer architecture with approximately 165M parameters. Following the masked diffusion approach, the model is trained to predict masked tokens in noisy text sequences, effectively learning a score function over text data. Full details are presented in Appendix B.

We use training set sizes $P \in \{64, 128, 256, 512, 1024\}$ ranging from 16,384 to 262,144 tokens. We track model performance using the validation loss on 19,531 sentences, which provides a lower bound to the negative log likelihood, and monitor memorization by generating 1,024 text samples at regular training checkpoints. Memorization is quantified by calculating the Hamming distance between each generated text sample and the closest training set text, averaged over the generations and divided by the sequence length. This metric captures the fraction of exact token matches between the generated and training text.

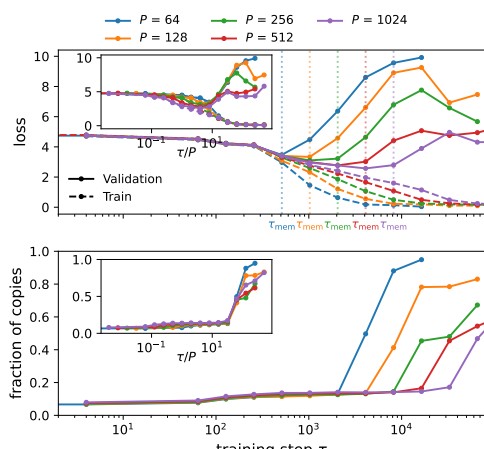

**Results and analysis** Figure 3 presents the results of this experiment. As with the vision diffusion models, MD4 initially generalizes, improving the log-likelihood on the validation corpus. However, after $\tau_{\text{mem}}$ the model begins to produce exact or near-exact copies of training text, signaling the onset of memorization. Notably, $\tau_{\text{mem}}$ scales linearly with the training set size $P$, consistent with our previous findings. The transition to memorization is also marked

Figure 3: **Memorization dynamics in language diffusion models.** Train loss, validation loss, and fraction of copied text as a function of training steps for GPT-based MD4 models trained on text8 with character-level tokenization and varying training set sizes $P$. Both losses initially decrease, indicating generalization, but diverge at the onset of memorization ($\tau_{\text{mem}}$), where the models start copying training text. $\tau_{\text{mem}}$ grows linearly with $P$ (insets).

by a sudden increase in the validation loss, indicating that early stopping can effectively prevent memorization also in this setting.

## 3.3 SUMMARY OF RESULTS

We have shown empirically that as they train, diffusion models generate higher and higher quality data, which are novel. This is true up to an early stopping time $\tau_{\text{mem}}$ where memorization starts, which we found to follow a remarkably universal empirical law:

$$\tau_{\text{mem}} \propto P. \tag{2}$$

**Theoretical support to the linear dependence** In Appendix G, we provide a theoretical basis for this scaling within the analytically tractable framework of kernel regression. We analyze the gradient flow dynamics for fitting the empirical score of $P$ training points in the low noise regime with variance $\sigma^2$, where the Gaussian modes centered at the training points are well separated. Using an ansatz for the score modes, we show that the time to fit the empirical score scales as $\tau_{\text{mem}} \propto P/\sigma^\nu$. The exponent $\nu$ is determined by the kernel's expansion near the origin. This result generalizes to any isotropic kernel the contemporaneous findings of Bonnaire et al. (2025), who studied random features in the proportional regime (width proportional to input dimension) using a Gaussian equivalence assumption. In particular, our results show that random features and neural networks in the Neural Tangent Kernel (NTK) regime (Jacot et al., 2018) have different behaviors.

We empirically validate these predictions with a one-hidden-layer network with lazy (NTK) initialization, trained by gradient descent to fit the empirical score of Gaussian random points. The observed $\tau_{\text{mem}}$ precisely follows the predicted scaling. Interestingly, the same scaling holds under feature learning initialization, suggesting our theory captures a more general phenomenon beyond its fixed-kernel assumption. Moreover, we show that $\tau_{\text{mem}}$ is insensitive to batch size – from small-batch SGD to full-batch – indicating that memorization time is governed by the number of optimization steps required to fit the empirical score, not by how often each example is revisited.

We will now study a controlled model of synthetic data that captures the phenomenology observed for natural data. Most importantly, it will allow us to quantify in detail the inaccuracy of generations of diffusion models with limited training, responsible for the inconsistent images in Figure 2.

## 4 Generalization vs. Memorization with a Simple Grammar

In this section, we consider diffusion models trained to generate sentences respecting the rules of a simple formal grammar.

### 4.1 Probabilistic graphical models

In theoretical linguistics, *Probabilistic Context-Free Grammars* (PCFG) have been proposed as a framework to describe the hierarchical structure of the syntax of several languages (Chomsky, 1956; Rozenberg & Salomaa, 1997; Pullum & Gazdar, 1982; Joshi, 1985; Manning & Schütze, 1999). Moreover, they have been proposed for describing semantic aspects of images under the name of *Pattern Theory* (Grenander, 1996; Jin & Geman, 2006; Siskind et al., 2007). PCFGs consist of a vocabulary of latent (*nonterminal*) symbols and a vocabulary of visible (*terminal*) symbols, together with probabilistic *production rules* establishing how one latent symbol generates tuples of symbols.

**The Random Hierarchy Model (RHM)** The RHM (Cagnetta et al., 2024) is a simple PCFG introduced as a theoretical toy model describing hierarchy and compositionality in data. With respect to generic PCFGs, it is built with some simplifying assumptions:

- Symbols are organized in a regular-tree topology of depth $L$ and branching factor $s$. The bottom layer, indexed as $\ell = 0$, corresponds to the leaves of the tree, which are the visible (terminal) symbols. The upper part of the tree, with layers $\ell = 1, \ldots, L$, corresponds to latent (nonterminal) symbols in the data structure.
- Nonterminal symbols are taken from $L$ finite vocabularies $(\mathcal{V}_\ell)_{\ell=1,\ldots,L}$ of size $v$ for each layer $\ell = 1, \ldots, L$. Terminal symbols belong to the vocabulary $\mathcal{V} \equiv \mathcal{V}_0$ of size $v$.
- The production rules transform one symbol in a node at level $\ell + 1$ into a string of $s$ symbols in its children nodes at level $\ell$. For each non-terminal symbol, there are $m$ rules with equal probability, which are *unambiguous*, i.e., two distinct symbols cannot generate the same $s$-string. Rules are sampled randomly without replacement and frozen for a given instance of the RHM. The $m$ strings generated by the same latent symbol are referred to as *synonyms*.

The fixed tree topology ensures that visible data at the leaves are strings of fixed length $d = s^L$. In analogy with language modeling, we call visible symbols *tokens*.

The number of possible data generated by this model is $vm^{\frac{d-1}{s-1}}$, which is exponential in the data dimension. Because of the random production rules, the tokens of the RHM data have non-trivial correlations reflecting the latent hierarchical structure (Cagnetta & Wyart, 2024).

### 4.2 Diffusion on the Random Hierarchy Model

**The exact score function of the RHM** Because of its correlations, the probability distribution of the RHM data and its corresponding score function are highly non-trivial. Nevertheless, if the production rules are known, thanks to the latent tree structure, the score function for any noise level can be computed exactly using the Belief Propagation (BP) algorithm. (Mezard & Montanari, 2009).

**Sample complexity** Favero et al. (2025) studied the sample complexity for diffusion models based on deep neural networks trained on finite RHM data. Their main findings are the following.

- The sample complexity to learn to generate valid data depends on the parameters of the model as $P^* \sim vm^{L+1}$, which is polynomial in the dimension, i.e., $P^* \sim vmd^{\log m / \log s}$. This scale can be theoretically predicted by comparing the size of the correlations between tokens and latent features, used in deep architectures for denoising, with their sampling noise.
- For $P < P^*$, there are regimes of partial generalization where the generated data are consistent with the rules up to layer $\ell$. The sample complexity to learn the rules at layer $\ell$ scales as $P_\ell^* \sim vm^{\ell+1}$.
- When $P > P_\ell^*$, the number of training steps $\tau_\ell^*$ required to learn the rules at layer $\ell$ is proportional to $P_\ell^*$, therefore having the same polynomial scaling with the dimension. Complete generalization is therefore achieved with $\tau^* \propto P^* = P_L^*$ number of training steps.

Notice that the sample complexity depends on the underlying distribution, e.g., the parameters of the grammar, and not on the specific number of available training samples.

### 4.3 GENERALIZATION VS. MEMORIZATION

We consider an instantiation of the RHM with a given set of parameters (depth $L$, branching factor $s$, vocabulary size $v$, and number of synonyms $m$). We generate $P$ distinct strings from this grammar, which constitute the training set. Each token is one-hot encoded, and we train a *Discrete Denoising Diffusion Probabilistic Model* (D3PM) (Austin et al., 2021) with uniform transition probabilities (Hoogeboom et al., 2021). The architecture of the diffusion model is made of a convolutional U-Net (Ronneberger et al., 2015) with $2L$ layers in total – $L$ in the encoder and $L$ in the decoder. We consider highly overparameterized networks with $8,192$ channels per layer, with a total number of parameters varying between $0.4$B for $L = 3$ and $0.7$B for $L = 5$. We use the maximal-update ($\mu$P) initialization to ensure feature learning (Yang & Hu, 2020). We train the neural network using Adam to optimize the training loss of discrete diffusion (Austin et al., 2021), derived from a variational bound on the negative log-likelihood (Sohl-Dickstein et al., 2015). Further experimental details are reported in Appendix B.

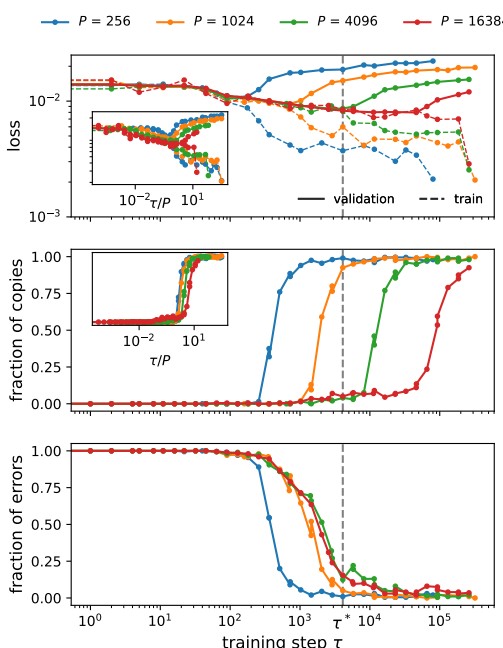

Figure 4: **Memorization vs. generalization on the RHM.** For training set size $P = 256$, the diffusion model generates valid data only when it is memorizing For $P = 16,384$, instead, the model generalizes, approximately at $\tau^*$, before starting to memorize. The memorization time scales linearly in $P$ (insets). Data for RHM parameters $v = 16$, $m = 4$, $L = 3$, $s = 2$.

We study the evolution of the models during training. For checkpoints at different training times, we track the training loss and the validation loss on $2,048$ held-out data. In addition, we generate $1,024$ data points with the diffusion model and measure their Hamming distance with the training data, determining if they are copies or not. We also check if the generated data are compatible with all the rules of the RHM, determining if they are valid strings of the grammar or not.

**Results and analysis** Figure 4 shows the evolution of a diffusion model during training with RHM parameters $v = 16$, $m = 4$, $L = 3$, $s = 2$. For these parameters, the sample complexity to learn all the rules of the grammar is $P^* \approx 4,096$. Varying the training set size $P$, we observe that the validation and training losses start decreasing at the same time and follow the same behavior until separating later in training, at a time depending on $P$. Comparing these losses with the fraction of copies between the generated data and the training ones, we observe that the increase of the validation loss corresponds to the onset of memorization. As observed for real data in section 3, we find

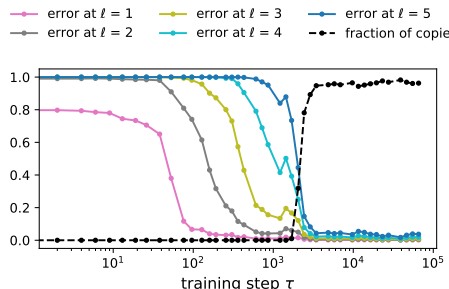
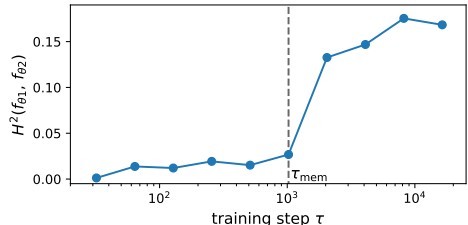

(a) Layer-wise learning in the RHM before memorization.

(b) Distance between the outputs of two diffusion models trained on disjoint training sets.

Figure 5: **Diffusion models achieve partial generalization in the RHM before memorizing.** (a) The diffusion model learns progressively deeper RHM rules during training. However, the rules at the deepest level $L = 5$ are never learned, and the corresponding error decreases only when memorization occurs, since $P = 1{,}024$ is smaller than the sample complexity $P_L^* \sim 10^4$. (b) Two diffusion models trained on disjoint training sets learn the same score function before the onset of memorization at $\tau_{mem}$. Data for RHM parameters $v = 16$, $m = 3$, $L = 5$, $s = 2$.

empirically that the onset of memorization requires a number of training steps $\tau_{\mathrm{mem}}$ proportional to $P$ (insets of Figure 4). The fraction of errors measures how many of the generated data are not compatible with the RHM rules. We observe that for $P < 4{,}096$, the fraction of errors decreases only in correspondence with memorization: the generated data are valid according to the grammar rules, but they are copies of the training set. For $P > 4{,}096$, instead, the fraction of errors decreases *before* the onset of memorization: the diffusion model is generating valid data that do not belong to the training set, and it is therefore generalizing. In Appendix F, we show that the generated data respect the correct statistics of the RHM rules, therefore learning the true data distribution. As a reference, Figure 4 reports the time $\tau^* = P^*$ as a vertical dashed line. We observe that the generalizing models ($P = 4{,}096$ and $P = 16{,}384$) achieve a fraction of errors $< 15\%$ for $\tau > \tau^*$. Therefore, these models present a dynamical phase $\tau^* < \tau < \tau_{\mathrm{mem}}$ where they achieve nearly perfect generalization before starting to memorize. This phase becomes longer with increasing $P$.

## 4.4 PARTIAL GENERALIZATION

For $P < P^*$, the diffusion model does not have enough training data to learn the deeper levels of the rules. However, it can still learn the lower levels of the rules up to layer $\tilde{\ell}$, with $P > P_{\tilde{\ell}}^*$, as the sample complexity $P_\ell^*$ increases with $\ell$. In this case, the model achieves *partial generalization*, corresponding to learning to generate data with some local coherence but lacking a global one, consistent with observations of Figure 2.

In Figure 5(a), a diffusion model is trained with $P = 1{,}024$ training points of an RHM with depth $L = 5$, while the sample complexity to learn all the rules is $P^* = P_L^* \simeq 10^4$. During training, we generate data with the diffusion model and measure if they are compatible with the RHM rules at layer $\ell$, measuring the corresponding fraction of errors. The figure shows that the errors at the layers $\ell \leq 3$ decrease at training times depending on $\ell$, in accordance with $\tau_\ell \propto P_\ell^*$ (Favero et al., 2025). However, for $\ell > 3$, the fractions of errors reach small values only at the onset of memorization $\tau_{\mathrm{mem}}$, when the fraction of copies of the training set goes up. This behavior implies that the model never learns the rules at the deeper levels $\ell = 4, 5$ since the number of training data is smaller than the sample complexity, and generates data with global consistency only when it starts memorizing.

**Even when partially generalizing, diffusion models learn the same score function** Even without achieving perfect generalization, diffusion models gradually improve their generalization during training – before memorizing – by capturing some structure of the underlying data distribution. In the RHM case, this corresponds to the lowest levels of the grammar. As a consequence, the score function that is learned during training *before memorization* is the same *independently* of the sampling of the training set. In Figure 5(b), we train two diffusion models in the same setting

as Figure 5(a) but with two disjoint training sets. We measure the difference in their outputs – i.e., the components of the learned score – during training by computing their Hellinger distance averaged over the tokens and the sampling of the diffusion trajectories from 1,024 test data. We observe that the distance between the output functions of the two models, i.e., the learned scores – which determine the generative process – remains stable during training and only jumps to higher values when the models start memorizing their respective training sets. Therefore, the two diffusion models learn very similar score functions when their generalization is gradually improving, before they overfit their respective empirical scores.

## 5 RELATED WORK

**Memorization in diffusion models**    Several works have documented the tendency of diffusion models to memorize the training data (Carlini et al., 2023; Somepalli et al., 2022; 2023; Wang et al., 2024). Dockhorn et al. (2022) proposes a mitigation strategy based on differentially private stochastic gradient descent, while Chen et al. (2024) introduces an anti-memorization guidance. Yoon et al. (2023); Kadkhodaie et al. (2023); Gu et al. (2025) interpret memorization as an overfitting phenomenon driven by the large capacity of overparameterized neural networks. Kadkhodaie et al. (2023) shows that underparameterized models trained on disjoint training sets learn the same score function, therefore generalizing by sampling the same target distribution; in contrast, overparameterized models memorize. Li et al. (2024); Wang & Vastola (2024) find that during their initial training phases, overparameterized diffusion models have an inductive bias towards learning a Gaussian approximation of data. This process achieves a primitive form of partial generalization by capturing some data's low-dimensional structure before the model begins to fully memorize the training points. Our results extend this viewpoint to later training stages and higher-order data statistics. Additionally, we quantify the timescale at which models transition from generalizing to memorizing.

**Overfitting in supervised learning vs. diffusion models**    Although the dynamics of first generalizing and then overfitting to the training data is observed also in some supervised learning settings (Advani et al., 2020; Nakkiran et al., 2021) – where recent theoretical progress has been made (Montanari & Urbani, 2025) – these problems have fundamental differences with memorization in diffusion models, i.e., learning the empirical score. For instance, in a typical regression task, a model fits a target function whose observations are assumed to be corrupted by external, unstructured noise. In the diffusion context, instead, the empirical score at low noise levels significantly differs from the population one: the corresponding "noise", i.e., the difference between the two functions, is inherent to the training set, structured, and defined over the entire domain of the inputs $x_t$. An overparameterized model converging to the empirical target, therefore, memorizes the training set and cannot generalize. This contrasts with noisy regression, where overparameterization can surprisingly be beneficial, leading to *double descent* (Spigler et al., 2019; Belkin et al., 2019) and *benign overfitting* (Bartlett et al., 2020).

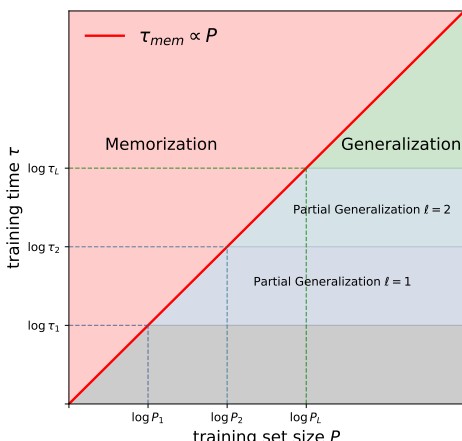

Figure 6: **Phase diagram of generalization vs. memorization** indicating different regimes as a function of training time $\tau$ and sample complexity $P$: partial generalization, (full) generalization and memorization. Note that in the RHM, learning proceeds through well-defined steps, while it is smoother for natural data.

## 6 CONCLUSION

We have argued that the learning dynamics in diffusion models is best understood as a competition between time scales, as summarized in Figure 6. A larger training set implies a larger memorization time, thus opening a larger time window to generate more coherent data. These results open new avenues for fine control of copyright issues, using early stopping to avoid memorization and building backward flows that are nearly independent of the training set, as we demonstrated.

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

## A   FURTHER RELATED WORK

**Theory of diffusion**   Under mild assumptions on the data distribution, diffusion models achieve a sample complexity scaling exponentially with data dimension (Block et al., 2020; Oko et al., 2023). The sampling and memorization process has been studied for Gaussian mixtures and linear manifolds using the empirical score function (Biroli et al., 2024; Ambrogioni, 2023; Achilli et al., 2024; 2025; Li & Chen, 2024). Learning the empirical score function was studied in (Cui et al., 2023; Shah et al., 2023; Han et al., 2024). The memorization-generalization trade-off in terms of model capacity with random features was studied in George et al. (2025). Generalization bounds for early-stopped random features learning simple score functions were derived in Li et al. (2023). (Biroli & Mézard, 2023; Ambrogioni, 2023; Biroli et al., 2024) show for Gaussian mixtures the existence of a characteristic noise level during the diffusion process where the single modes merge into one. In Biroli et al. (2024), another noise scale is identified, corresponding to short diffusion times, where the backward process collapses into the single training data points, associated with memorization. Kamb & Ganguli (2024) studies generalization in vision diffusion models through the inductive bias of translational equivariance and locality.

**Diffusion models for hierarchical data**   For hierarchically structured data, Sclocchi et al. (2024b;a) show that the reconstruction of high-level features undergoes a phase transition in the diffusion process, while low-level features vary smoothly around the same noise scale. For the same data model, Favero et al. (2025) shows that U-Net diffusion models learn to generate these data by sequentially learning different levels of the grammatical rules, with a sample complexity polynomial in data dimension. Sclocchi et al. (2024b) shows that the Bayes-optimal denoising algorithm for hierarchical data corresponds to belief propagation; Mei (2024) shows that U-Net architectures are able to efficiently approximate this algorithm. Moreover, Garnier-Brun et al. (2024) shows that transformers can implement the same algorithm.

## B   EXPERIMENTAL DETAILS

### B.1   VISION DIFFUSION MODELS

**iDDPM**   In our experiments, we utilize Improved Denoising Diffusion Probabilistic Models (iDDPMs) for image generation on the CIFAR-10 and CelebA datasets, following the codebase of Improved DDPMs (Nichol & Dhariwal, 2021): `https://github.com/openai/improved-diffusion`. Specifically, we train iDDPMs with 256 and 128 channels for CIFAR-10 and CelebA, respectively. Our models are implemented using a U-Net architecture with attention layers and 3 resolution blocks. We use $4,000$ diffusion steps, a cosine noise schedule, a learning rate of $10^{-4}$, and a batch size of $128$. Training is performed for $262,144$ steps using a *hybrid objective* (Nichol & Dhariwal, 2021) and the Adam optimizer with dropout of $0.3$.

**Stable Diffusion**   We fine-tune Stable Diffusion v2.1[1] using the codebase `https://github.com/somepago/DCR` from Somepalli et al. (2022; 2023). The model is pre-trained on LAION-2B (Schuhmann et al., 2022) and consists of a latent diffusion U-Net architecture with frozen text and autoencoder components. We fine-tune the U-Net for $262,144$ steps on $8,192$ images from the LAION-10k dataset at resolution $256 \times 256$, using a batch size of $16$. We employ a constant learning rate of $5 \times 10^{-6}$ with 5,000 warm-up steps and use a single image-caption pair per datapoint.

### B.2   LANGUAGE DIFFUSION MODELS

**MD4**   Our experiments leverage the codebase of MD4 (Shi et al., 2024), available at `https://github.com/google-deepmind/md4`. MD4 is a masked diffusion model that progressively transforms tokens into a special [MASK] token as training proceeds. Specifically, at each timestep $t$, each non-masked token has a probability $\beta_t$ of being replaced by [MASK]. The forward transition process for this model can be formally described using a one-hot encoding of the $|\mathcal{V}| + 1$ states, where the transition matrix is defined as:

$$Q_t = (1 - \beta_t)\mathbb{I} + \beta_t \mathbf{1}\mathbf{e}_M^\top. \tag{3}$$

---

[1] `https://huggingface.co/stabilityai/stable-diffusion-2-1`

Here $\mathbb{I}$ the identity matrix, $\mathbf{1}$ a vector of ones and $\mathbf{e}_M$ the one-hot-encoding vector corresponding to the [MASK] symbol. The entries $[Q_t]_{ij}$ of $Q_t$ indicate the probability of the token $x_k$ transitioning from state $i$ to state $j$, i.e., $[Q_t]_{ij} = q(x_{k,t} = j | x_{k,t-1} = i)$. At the final timestep $T$, all tokens are fully masked, i.e., $x_{k,T} = $ [MASK] for every $k \in [\dim(x)]$. For our experiments, we train MD4 using a batch size of $64$ and a context size of $256$. All other hyperparameters are kept consistent with the original MD4 implementation.

### B.3 RANDOM HIERARCHY MODEL

**D3PM** For our experiments on the Random Hierarchy Model, we employ convolutional U-Net-based Discrete Denoising Diffusion Probabilistic Models (D3PMs) (Austin et al., 2021). These models are tasked to predict the conditional expectation $\mathbb{E}(x_0 | x_t)$, which parameterizes the reverse diffusion process. In particular, we consider a uniform diffusion process (Hoogeboom et al., 2021; Austin et al., 2021), where, at each timestep $t$, tokens can either stay unchanged or, with probability $\beta_t$, can transition to some other symbol in the vocabulary. One-hot encoding the $|\mathcal{V}|$ states, the forward transition matrix formally reads:

$$Q_t = (1 - \beta_t)\mathbb{I} + \frac{\beta_t}{|\mathcal{V}|}\mathbf{1}\mathbf{1}^\top. \tag{4}$$

Here $\mathbb{I}$ is the identity and $\mathbf{1}$ is a vector of all ones. At the final time $T$, the stationary distribution is uniform over the vocabulary. The convolutional U-Net has $L$ resolution blocks in both the encoder and decoder parts. Each block features the following specification: filter size $s$, stride $s$, 8,192 channels per layer, GeLU non-linearity, skip connections linking encoder and decoder blocks of matching resolution to preserve multi-scale feature information. We include embedding and un-embedding layers implemented as convolutional layers with a filter size of $1$. This architecture is specifically aligned with the RHM's hierarchical structure, where the filter size and stride of $s$ in the convolutional layers mirror the branching factor of the RHM tree. While this design provides practical benefits in terms of training efficiency, it should not alter the fundamental sample complexity of the problem, as long as the network is sufficiently deep and expressive (Cagnetta et al., 2024). The networks are initialized with the maximal-update ($\mu$P) parameterization (Yang & Hu, 2020), ensuring stable feature learning even in the large-width regime. We train with Adam with a learning rate of $0.1$ and a batch size of $32$. For the diffusion process, we adopt a linear schedule with $1,000$ noise levels.

### B.4 HARDWARE

All experiments are run on a single NVIDIA H100 SXM5 GPU with 94GB of RAM.

## C EXPERIMENTS ON STABLE DIFFUSION

We consider Stable Diffusion v2.1 (Ronneberger et al., 2015), a text-to-image latent diffusion model pre-trained on the LAION-2B dataset (Schuhmann et al., 2022). We fine-tune this model for $262,144$ steps on $8,192$ samples from the LAION-10k dataset (Somepalli et al., 2023), using a resolution of $256 \times 256$. During fine-tuning, the text encoder and encoder-decoder components are kept frozen. We use a held-out validation set of $1,024$ image-text pairs to monitor the validation loss. Full training details are provided in Appendix B.

To quantify memorization, we follow the protocol of Somepalli et al. (2022) and compute a similarity score for each generated image based on the cosine similarity of SSCD (Self-Supervised Descriptor for Image Copy Detection) (Pizzi et al., 2022) features, extracted from a ResNet-50 model. Each score is defined as the similarity between a generated image and its nearest neighbor in the training set.

Figure 7(a) plots the training and validation losses as a function of the training step $\tau$. As observed in the main text, initially, both losses decrease, indicating generalization: the model output aligns increasingly with the population score. At a critical time $\tau_{\mathrm{mem}} \propto P$, the validation loss diverges from the training loss, marking the onset of memorization. Early stopping at this point can prevent the model from entering the memorization phase.

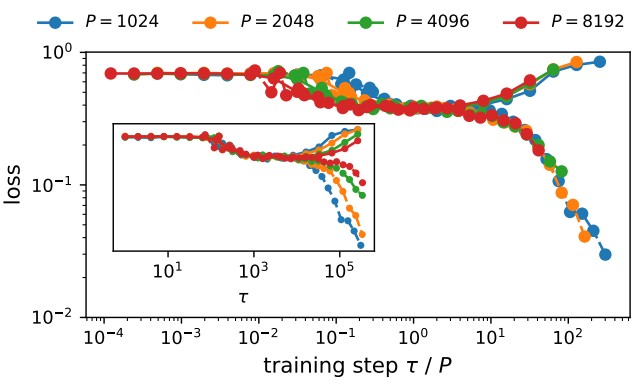
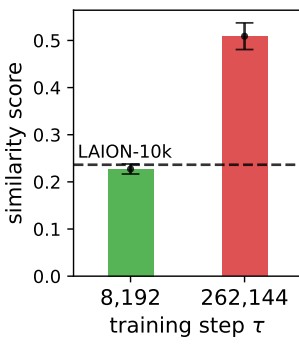

(a) Generalization-memorization dynamics.

(b) Similarity scores.

Figure 7: **Memorization dynamics in Stable Diffusion.** (a) Training and validation losses as a function of training step $\tau$ for Stable Diffusion fine-tuned on different subset of LAION-10k with $P$ training points. Both losses initially decrease, indicating generalization, and diverge at the memorization onset time $\tau_{\mathrm{mem}}$. The memorization time $\tau_{\mathrm{mem}}$ is linear in the training set size $P$. (b) Cosine similarity scores between SSCD ResNet embedding for generated images and their nearest training neighbor at early stopping ($\tau = 8{,}192$) and final training ($\tau = 262{,}144$). The dashed line indicates the mean similarity score between the closest LAION-10k samples. The sharp increase at late training signals memorization.

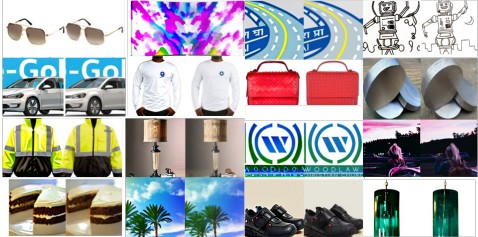

Figure 8: **Replicates generated by Stable Diffusion.** Example generations (left) from the final training checkpoint ($\tau = 262{,}144$) with similarity score $> 0.5$ to their nearest neighbor in the training set (right), confirming memorization.

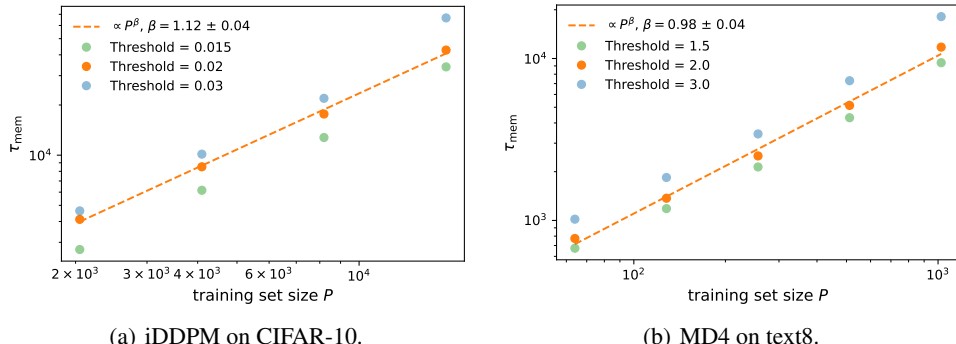

(a) iDDPM on CIFAR-10.

(b) MD4 on text8.

Figure 9: **Scaling of memorization time with dataset size.** For each dataset size $P$, we linearly interpolate the train and validation losses as a function of the training step $\tau$ (on a logarithmic grid) and define the memorization time $\tau_{\mathrm{mem}}$ as the first step at which the interpolated loss gap $L_{\mathrm{val}}(\tau) - L_{\mathrm{train}}(\tau)$ exceeds a fixed threshold (different colors). We then plot $\tau_{\mathrm{mem}}$ as a function of $P$ for (a) iDDPMs trained on CIFAR-10 and (b) MD4 language diffusion models trained on text8. In both cases, the data are well described by a power-law fit $\tau_{\mathrm{mem}} \propto P^{\beta}$ (dashed lines) with exponents $\beta$ close to one, indicating an approximately linear growth of memorization time with dataset size across modalities.

In Figure 7(b), we report the similarity scores for 200 generated images at two checkpoints: early stopping ($\tau = 8{,}192$) and the final training step ($\tau = 262{,}144$). For reference, we also show the similarity score for real images from the full LAION-10k dataset (black dashed line). At the early stopping time, the generated images exhibit diversity similar to that of the dataset. In contrast, by the end of training, the similarity score increases by a factor of two, indicating memorization.

Finally, in Figure 8, we show representative examples of replicated samples (similarity score $> 0.5$) from the final checkpoint, confirming that Stable Diffusion memorized part of its training set.

## D  FURTHER RESULTS ON IDDPMS

**Scaling of memorization time**  To further quantify how memorization time scales with dataset size, we estimate a memorization onset time $\tau_{\mathrm{mem}}(P)$ for each number of training examples $P$. For every setting of $P$, we record the training and validation losses as a function of the training step $\tau$ and linearly interpolate them in $\tau$ on a logarithmic grid to obtain dense loss curves. We then consider the difference between validation and training loss and define $\tau_{\mathrm{mem}}(P)$ as the first training time at which this loss gap exceeds a fixed threshold value. The resulting $\tau_{\mathrm{mem}}$ values for iDDPMs on CIFAR-10 are shown in 9(a), where they are well described by a power-law fit $\tau_{\mathrm{mem}} \propto P^{\beta}$ with $\beta \approx 1$, indicating an approximately linear growth of memorization time with dataset size.

**FID dynamics**  Figure 10 reports the Fréchet Inception Distance (FID) as a function of the training step $\tau$ for a DDPM trained on $16{,}384$ CIFAR-10 images, consistent with the setup in Figure 1. At each checkpoint, we generate $32{,}768$ samples and compute the FID against the union of CIFAR-10 standard train and test splits. The FID captures both the quality and diversity of the generated images. As training progresses, the FID decreases monotonically until the memorization onset time $\tau_{\mathrm{mem}}$, after which it gradually increases – reflecting a loss in sample diversity as the model begins replicating its training data.

**Further examples of generations**  Figure 11 presents further images sampled from the early stopped iDDPM trained on $16{,}384$ CIFAR-10 images.

**Examples of copies**  Figure 12 shows examples of generated samples (top row) and their nearest neighbors in the training set (bottom row) for the iDDPM trained on $8{,}192$ CIFAR-10 images. These examples are taken from the end of training, within the memorization phase, where the model begins to replicate its training data.

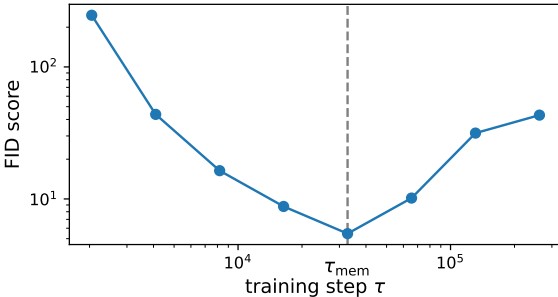

Figure 10: **FID dynamics.** Fréchet Inception Distance (FID) as a function of training step $\tau$ for a DDPM trained on 16,384 CIFAR-10 images. The FID initially decreases, reflecting improved generation quality and diversity, but begins to rise past $\tau_{\mathrm{mem}}$ as the model starts copying training examples.

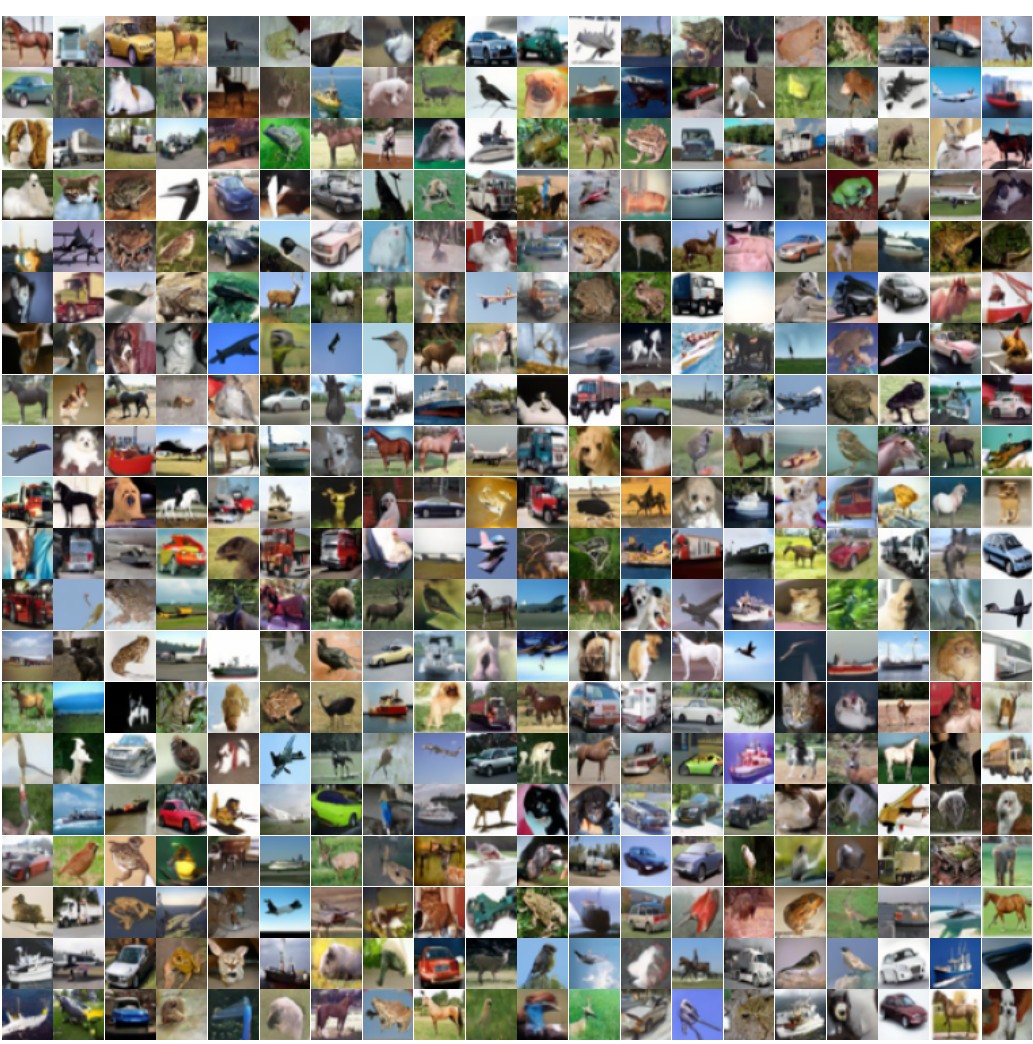

Figure 11: **CIFAR-10 samples generated with early-stopped model.** Additional samples from the iDDPM trained on 16,384 CIFAR-10 images, generated at the early stopping point before memorization. The model produces diverse and high-quality images without replicating the training data.

Figure 12: **Examples of copies on CIFAR-10.** Top: samples generated by the iDDPM trained on 8,192 CIFAR-10 images at the end of training. Bottom: nearest neighbors from the training set. The model reproduces specific training examples, indicating memorization.

**Generated sample 1**

on french anarchism reemerged
influencing the bourses de
travails af autonomous workers
groups and trade unions from
this movement the conf d ration
g n rale du travail general
confederation of work cgt was
formed in one eight nine flve as
the first major

**Nearest training sequence**

on french anarchism reemerged
influencing the bourses de
travails of autonomous workers
groups and trade unions from
this movement the conf d ration
g n rale du travail general
confederation of work cgt was
formed in one eight nine five as
the first major

**Generated sample 2**

erpetual covenant of wareagayndt
every people tribe and state
ocning a foot of land between
here and tierra dul fuego beeause
opposition to slavery expansion
was the key issue uniting the
redublican patty at the time
lincol is sometimes critic zed
for put

**Nearest training sequence**

erpetual covenant of war against
every people tribe and state
owning a foot of land between
here and tierra del fuego because
opposition to slavery expansion
was the key issue uniting the
republican party at the time
lincoln is sometimes criticized
for put

**Generated sample 3**

s with a hagh levul tf
intellectual function ne have
aspirger s autasm or taat both
lypes are merely g eks with a
medical label attacied also
ausicm has evolved in the
public understantingpbot the
populariidentification of
autism with relatinely severe
cas

**Nearest training sequence**

s with a high level of
intellectual functioning have
asperger s autism or that both
types are merely geeks with a
medical label attached also
autism has evolved in the
public understanding but the
popular identification of
autism with relatinely severe
cas

Figure 13: **Examples of copies on text8.** Left: Diffusion-generated text with MD4 trained on $1,024$ training sequences of the text8 dataset for $524,288$ SGD steps. Right: the corresponding nearest training sequences. The generated samples are copies of the training set, up to a few character-level mistakes.

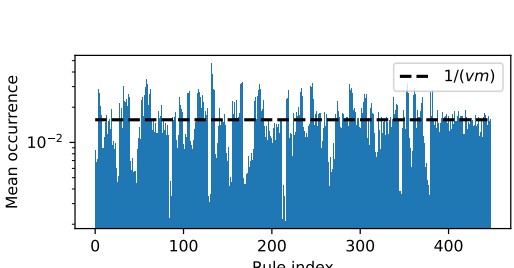 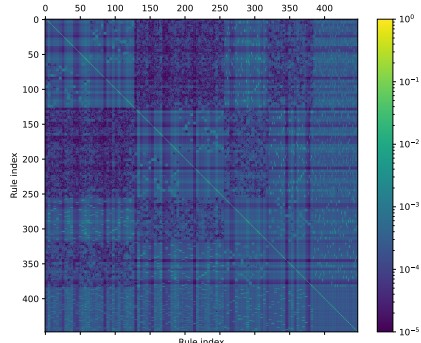

Figure 14: **Sampling of RHM production rules.** Mean occurrence (*left*) and centered covariance (*right*) of the production rules sampled by a diffusion model trained on $P = 16{,}384$ strings ($v = 16$, $m = 4$, $L = 3$, $s = 2$). The model, trained with early stopping ($\tau = 32{,}768$), samples all RHM rules with a mean occurrence that is approximately uniform (up to sampling noise). Likewise, the correlations between the cooccurrence of sampled rules show that they are sampled approximately independently.

## E   FURTHER RESULTS ON MD4

**Scaling of memorization time**   Similarly to DDPMs, in Figure 9(b) we study how the memorization time scales with the number of training examples $P$ for GPT-based MD4 models trained on text8. The resulting $\tau_{\text{mem}}$ values exhibit a clear power-law dependence on $P$, $\tau_{\text{mem}} \propto P^{\beta}$, with an exponent $\beta$ close to one, mirroring the behavior observed for iDDPMs and indicating a similar linear growth of memorization time with dataset size in the language setting.

**Examples of copies**   Figure 13 shows examples of generated samples and their nearest neighbors in the training set for the MD4 trained on 1,024 text8 sequences. These examples are taken from the end of training, within the memorization phase, where the model begins to replicate its training data. In particular, the generated samples are copies of the training set, up to a few character-level mistakes.

## F   FURTHER RESULTS ON THE RHM

**Production rules sampling**   Figure 14 shows the mean occurrence and centered covariance of the production rules sampled by a diffusion model trained on $P = 16{,}384$ strings ($v = 16$, $m = 4$, $L = 3$, $s = 2$). The model, trained with early stopping ($\tau = 32{,}768$), samples all RHM rules with a mean occurrence that is approximately uniform (up to sampling noise); likewise, the correlations between the cooccurrence of sampled rules show that they are sampled approximately independently. Therefore, the generated data reproduces the correct data distribution of the RHM, corresponding to generalization.

**Robustness to optimizer choice**   To test robustness to the optimization dynamics, we repeated the RHM experiments with $L = 3$, $s = 3$, $v = 24$, $m = 12$ using the same D3PM architecture and training protocol as in the main text, varying only the optimizer between Adam and SGD. For each optimizer, we set the learning rate to its *maximal stable* learning rate, defined as the largest value for which the training loss reliably converges. Figure 15 confirms the robustness of $\tau_{\text{mem}}$ to the choice of optimizer in this setting.

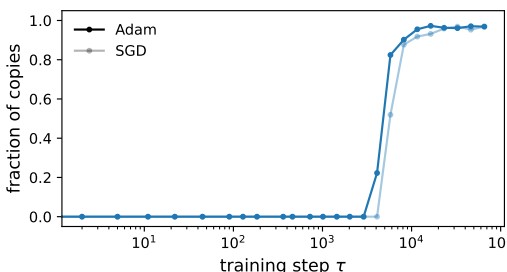

Figure 15: **Effect of the optimizer on the onset of memorization in the RHM.** Fraction of copies as a function of training step for a D3PM trained on $P = 2,048$ sequences sampled from an RHM with $L = 3$, $s = 3$, $v = 24$, $m = 12$, using Adam and SGD, each with its maximal stable learning rate (largest learning rate yielding convergent training loss). The curves nearly coincide, and the onset of memorization occurs at approximately the same number of training steps for both optimizers.

## G    SCALING ARGUMENT FOR THE MEMORIZATION TIME OF KERNEL METHODS

In this section, we analyze the training time $\tau_{\mathrm{mem}}$ required for a kernel to learn the score of $P$ well-separated training points in the low-noise limit for a fixed noise level. This timescale corresponds to the one for diffusion models to memorize the training data.

**Setting**    We assume the empirical data distribution is the Gaussian mixture

$$p_\sigma(x) = \frac{1}{P} \sum_{j=1}^{P} \mathcal{N}(x|x_j,\, \sigma^2 \mathbb{I}_d), \tag{5}$$

where the $x_j \in \mathbb{R}^d$ are $P$ distinct training points. We work in a low-noise limit, where the noise standard deviation $\sigma$ is much smaller than the typical distance between data points, i.e., $\sigma \ll \min_{j \neq i} \|x_i - x_j\|$. This ensures that the Gaussian components have negligible overlap, so $p_\sigma$ is approximately supported on $P$ disjoint neighborhoods.

We consider learning the score $\nabla_x \log p_\sigma(x)$ at fixed $\sigma$ with kernel regression. The dynamics of learning is governed by the spectral properties of the integral operator of the kernel $K$, defined as

$$(Kf)(x) = \int K(x, y) f(y) dp_\sigma(y), \tag{6}$$

with respect to the measure $p_\sigma$. The learning time for a specific mode (eigenfunction) of the data scales inversely with the corresponding eigenvalue of this operator.

We assume that the kernel $K(x, y)$ can be expanded for small distances $r = \|x - y\|$ as

$$K(x, y) = \kappa(r) = 1 - C(d)\, r^\nu + \mathcal{O}(r^{\nu+1}) \quad \text{as } r \to 0, \tag{7}$$

with $C(d)$ a coefficient that depends on the choice of the kernel and the input dimension $d$. For instance, the Neural Tangent Kernel (NTK) (Jacot et al., 2018) of neural networks with ReLU activations corresponds to $\nu = 1$, while their Random Feature Kernel (RFK) corresponds to $\nu = 2$.

**Local eigenfunctions**    In the low-noise limit, the score in the vicinity of a data point $x_i$ is dominated by the $i$-th Gaussian component:

$$\nabla_x \log p_\sigma(x) \simeq \nabla_x \log \left[ \frac{1}{P} \mathcal{N}(x|x_i,\, \sigma^2 \mathbb{I}_d) \right] = -\frac{x - x_i}{\sigma^2}. \tag{8}$$

This shows that the target function is locally linear and motivates our ansatz of approximate eigenfunctions to probe the spectrum of $K$. In particular, we construct a set of vector-valued functions

$\{\psi_i\}_{i \in [P]}$ centered at each data point $x_i$:

$$\psi_i(x) = (x - x_i) R\left(\frac{\|x - x_i\|}{\sigma}\right), \tag{9}$$

where $R : [0, \infty) \to \mathbb{R}$ is a smooth cutoff function (e.g., $R(r) = e^{-r}$) that decays rapidly for $r \gtrsim 1$. The support of $\psi_i$ is thus concentrated in the ball $B_\sigma(x_i)$. These functions are asymptotically orthogonal in $L_2(p_\sigma)$: $\langle \psi_i, \psi_j \rangle_{L_2(p_\sigma)} = \mathcal{O}(e^{-c/\sigma^2})$ for $i \neq j$.

**Eigenvalues and memorization time**   We compute the eigenvalue $\lambda_i$ associated with each $\psi_i$:

$$\lambda_i = \frac{\langle \psi_i, K\psi_i \rangle_{L_2(p_\sigma)}}{\|\psi_i\|^2_{L_2(p_\sigma)}}. \tag{10}$$

The squared norm is dominated by the integral over the $i$-th component of the mixture:

$$\|\psi_i\|^2_{L_2(p_\sigma)} = \int \|\psi_i(x)\|^2 p_\sigma(x) d^d x \simeq \frac{1}{P} \int \|x - x_i\|^2 R^2\left(\frac{\|x - x_i\|}{\sigma}\right) \mathcal{N}(x|x_i, \sigma^2 \mathbb{I}_d) d^d x. \tag{11}$$

Changing to local coordinates $u = \frac{x - x_i}{\sigma}$:

$$\|\psi_i\|^2_{L_2(p_\sigma)} \simeq \frac{\sigma^2}{P} \int \|u\|^2 R^2(\|u\|) \mathcal{N}(u|0, \mathbb{I}_d) d^d u \tag{12}$$

$$:= \gamma_d(R) \frac{\sigma^2}{P}, \tag{13}$$

where

$$\gamma_d(R) = \int \|u\|^2 R^2(\|u\|) \mathcal{N}(u|0, \mathbb{I}_d) d^d u \tag{14}$$

is a dimension-dependent constant (for fixed $R$) that does not depend on $\sigma$ or $P$.

The numerator is given by the quadratic form

$$\langle \psi_i, K\psi_i \rangle_{L_2(p_\sigma)} = \iint \psi_i(x) \cdot \psi_i(y) K(x, y) p_\sigma(x) p_\sigma(y) d^d x \, d^d y. \tag{15}$$

Given the localized support of $\psi_i$ and the non-overlapping assumption for the Gaussians, the integral is non-negligible only when both $x$ and $y$ are near $x_i$:

$$\langle \psi_i, K\psi_i \rangle_{L_2(p_\sigma)} \simeq \frac{1}{P^2} \iint \psi_i(x) \cdot \psi_i(y) K(x, y) \mathcal{N}(x|x_i, \sigma^2 \mathbb{I}_d) \mathcal{N}(y|x_i, \sigma^2 \mathbb{I}_d) d^d x \, d^d y. \tag{16}$$

We now substitute the expansion of the kernel near the origin:

$$\langle \psi_i, K\psi_i \rangle_{L_2(p_\sigma)} \simeq \frac{1}{P^2} \left[\int \psi_i(x) \mathcal{N}(x|x_i, \sigma^2 \mathbb{I}_d) d^d x\right] \cdot \left[\int \psi_i(y) \mathcal{N}(y|x_i, \sigma^2 \mathbb{I}_d) d^d y\right] \tag{17}$$

$$- \frac{C(d)}{P^2} \iint \psi_i(x) \cdot \psi_i(y) \|x - y\|^\nu \mathcal{N}(x|x_i, \sigma^2 \mathbb{I}_d) \mathcal{N}(y|x_i, \sigma^2 \mathbb{I}_d) d^d x \, d^d y.$$

The first term vanishes because $\psi_i(x)$ is an odd function with respect to the center $x_i$, while $\mathcal{N}(x|x_i, \sigma^2 \mathbb{I}_d)$ is even. The integral is therefore zero. The leading contribution comes from the second term. We again change variables to $u = (x - x_i)/\sigma$ and $v = (y - x_i)/\sigma$ obtaining

$$\langle \psi_i, K\psi_i \rangle_{L_2(p_\sigma)} \simeq -\frac{C(d)}{P^2} \iint [\sigma u R(\|u\|)] \cdot [\sigma v R(\|v\|)] (\sigma \|u - v\|)^\nu \mathcal{N}(u|0, \mathbb{I}_d) \mathcal{N}(v|0, \mathbb{I}_d) d^d u \, d^d v \tag{18}$$

$$= -C(d) \frac{\sigma^{2+\nu}}{P^2} \iint (u \cdot v) R(\|u\|) R(\|v\|) \|u - v\|^\nu \mathcal{N}(u|0, \mathbb{I}_d) \mathcal{N}(v|0, \mathbb{I}_d) d^d u \, d^d v. \tag{19}$$

We denote the remaining integral by

$$\beta_d(R, \nu) := \iint (u \cdot v) R(\|u\|) R(\|v\|) \|u - v\|^\nu \mathcal{N}(u|0, \mathbb{I}_d) \mathcal{N}(v|0, \mathbb{I}_d) d^d u \, d^d v, \tag{20}$$

so that

$$\langle \psi_i, K\psi_i \rangle_{L_2(p_\sigma)} \simeq -C(d)\,\beta_d(R,\nu)\,\frac{\sigma^{2+\nu}}{P^2}. \tag{21}$$

Collecting everything, the eigenvalue is

$$\lambda_i = \frac{\langle \psi_i, K\psi_i \rangle_{L_2(p_\sigma)}}{\|\psi_i\|^2_{L_2(p_\sigma)}} \;\simeq\; -C(d)\,\frac{\beta_d(R,\nu)}{\gamma_d(R)}\,\frac{\sigma^{2+\nu}/P^2}{\sigma^2/P} \;=\; -C(d)\,\frac{\beta_d(R,\nu)}{\gamma_d(R)}\,\frac{\sigma^\nu}{P}. \tag{22}$$

Thus, up to a dimension- and kernel-dependent prefactor $-C(d)\,\beta_d(R,\nu)/\gamma_d(R)$, we obtain

$$\lambda_i \propto \frac{\sigma^\nu}{P}. \tag{23}$$

The training time required to learn these localized eigenfunction scales as the inverse of the eigenvalue. This defines the memorization timescale

$$\tau_{\mathrm{mem}} \sim \lambda_i^{-1} \sim \frac{P}{\sigma^\nu}. \tag{24}$$

**Dimension dependence of $\gamma_d$ and $\beta_d$.** The constants $\gamma_d(R)$ and $\beta_d(R,\nu)$ depend only on $d$, $\nu$, and the choice of cutoff $R$. To make their $d$-dependence explicit, it is convenient to specialize to the simplest case $R \equiv 1$. This simplification is justified because the factors $\mathcal{N}(u|0,\mathbb{I}_d)$ and $\mathcal{N}(v|0,\mathbb{I}_d)$ already suppress the integrand exponentially for $\|u\| \gg 1$ or $\|v\| \gg 1$.

In that case,

$$\gamma_d(R \equiv 1) = \int \|u\|^2\,\mathcal{N}(u \mid 0, \mathbb{I}_d)\,d^d u = \mathbb{E}[\|u\|^2] = d, \tag{25}$$

so

$$\gamma_d(1) = d. \tag{26}$$

For the numerator constant,

$$\beta_d(1,\nu) = \iint (u \cdot v)\,\|u - v\|^\nu\,\mathcal{N}(u \mid 0, \mathbb{I}_d)\,\mathcal{N}(v \mid 0, \mathbb{I}_d)\,d^d u\,d^d v = \mathbb{E}\big[u \cdot v\,\|u - v\|^\nu\big], \tag{27}$$

with $u, v \overset{\text{i.i.d.}}{\sim} \mathcal{N}(0, \mathbb{I}_d)$. Introducing the variables

$$a = \frac{u + v}{\sqrt{2}}, \qquad b = \frac{u - v}{\sqrt{2}}, \tag{28}$$

we have $a, b \sim \mathcal{N}(0, \mathbb{I}_d)$ independent, and

$$u \cdot v = \frac{1}{2}\big(\|a\|^2 - \|b\|^2\big), \qquad \|u - v\| = \sqrt{2}\,\|b\|. \tag{29}$$

Therefore

$$\beta_d(1,\nu) = \mathbb{E}\big[u \cdot v\,\|u - v\|^\nu\big] = \mathbb{E}\left[\frac{1}{2}\big(\|a\|^2 - \|b\|^2\big)\,(\sqrt{2}\|b\|)^\nu\right]. \tag{30}$$

Using the independence of $a$ and $b$ and the solution for the radial moments of isotropic Gaussians,

$$\beta_d(1,\nu) = \frac{1}{2}\left(\mathbb{E}[\|a\|^2]\,\mathbb{E}[(\sqrt{2}\|b\|)^\nu] - \mathbb{E}\big[\|b\|^2(\sqrt{2}\|b\|)^\nu\big]\right) \tag{31}$$

$$= \frac{1}{2}\left(d\,2^\nu\,\frac{\Gamma\left(\frac{d+\nu}{2}\right)}{\Gamma\left(\frac{d}{2}\right)} - 2^{\nu+1}\,\frac{\Gamma\left(\frac{d+\nu+2}{2}\right)}{\Gamma\left(\frac{d}{2}\right)}\right) \tag{32}$$

$$= \frac{1}{2}2^\nu\,\frac{1}{\Gamma\left(\frac{d}{2}\right)}\left(d\,\Gamma\left(\frac{d+\nu}{2}\right) - 2\,\Gamma\left(\frac{d+\nu+2}{2}\right)\right) \tag{33}$$

$$= -\frac{\nu}{2}\,2^\nu\,\frac{\Gamma\left(\frac{d+\nu}{2}\right)}{\Gamma\left(\frac{d}{2}\right)}. \tag{34}$$

Thus, for $R \equiv 1$ the ratio appearing in the eigenvalue is

$$\frac{\beta_d(1,\nu)}{\gamma_d(1)} = -\frac{\nu}{2}\, 2^\nu\, \frac{1}{d}\, \frac{\Gamma\!\left(\frac{d+\nu}{2}\right)}{\Gamma\!\left(\frac{d}{2}\right)}. \tag{35}$$

Using the large-$d$ asymptotics of the Gamma function,

$$\frac{\Gamma\!\left(\frac{d+\nu}{2}\right)}{\Gamma\!\left(\frac{d}{2}\right)} \sim \left(\frac{d}{2}\right)^{\nu/2}, \qquad d \to \infty, \tag{36}$$

we obtain

$$\frac{\beta_d(1,\nu)}{\gamma_d(1)} \sim -2^{\nu/2-1}\,\nu\, d^{\nu/2-1}. \tag{37}$$

Plugging this into the eigenvalue expression

$$\lambda_i \simeq -C(d)\,\frac{\beta_d(1,\nu)}{\gamma_d(1)}\,\frac{\sigma^\nu}{P} \sim 2^{\nu/2-1}\,\nu\, C(d)\, d^{\nu/2-1}\,\frac{\sigma^\nu}{P}. \tag{38}$$

Consequently, the memorization time scales as

$$\tau_{\mathrm{mem}} \sim \lambda_i^{-1} \sim C(d)^{-1}\, d^{1-\nu/2}\,\frac{P}{\sigma^\nu}. \tag{39}$$

**Remarks** Notice that this result is distribution-agnostic as it simply uses the fact that training points are isolated. Moreover, as long as diffusion happens in the ambient space, the same argument applies to data supported on a manifold of lower dimension $d_{\mathrm{eff}}$, so the intrinsic dimension of the data does not affect our results, i.e., Equation 39 depends on $d$ only and not on $d_{\mathrm{eff}}$.

All in all, this argument extends the results from contemporaneous work on random features in the proportional regime (number of neurons proportional to the input dimension) (Bonnaire et al., 2025) to any isotropic kernels. Our derivation relies only on the local behavior of the kernel and shows that random features and neural networks in the NTK limit have different behaviors.

**Numerical experiments** We confirm our theoretical scaling numerically in Figure 16 for a one-hidden-layer fully-connected network in the lazy (NTK) regime (Chizat et al., 2019). Notably, the same experimental setting under a mean-field (feature learning) initialization (Mei et al., 2018) also exhibits a memorization time consistent with our NTK-based prediction.

Furthermore, Figure 18 investigates the effect of batch size $B$. For both lazy and feature learning regimes, the timescale to fit the empirical score appears independent of $B$, from small-batch SGD ($B = 8$) to full-batch gradient descent ($B = P$). This observation implies that the memorization time only depends on the size of the training set and not on the number of times a training point is observed.

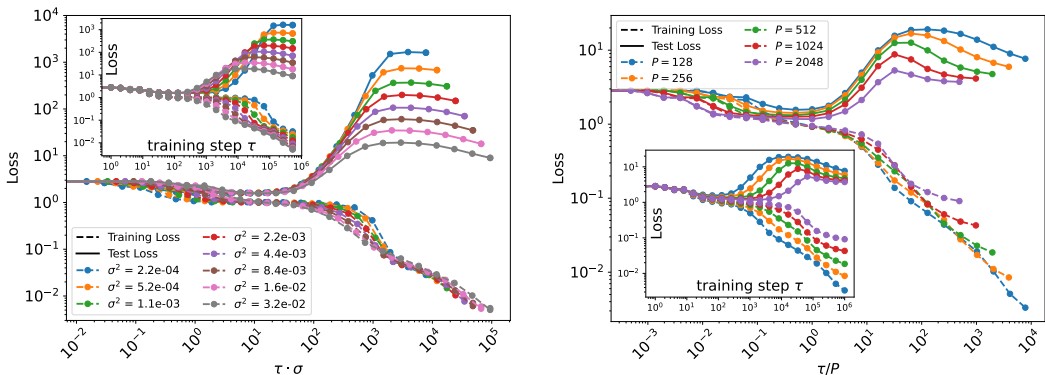

Figure 16: **Neural Tangent Kernel (NTK) initialization: one-hidden layer ReLU neural network (width $8192$) learning the empirical score at fixed diffusion noise variance $\sigma^2$, trained with full-batch gradient descent.** Training points sampled from a Gaussian distribution in $d = 64$ dimensions. *Left*: at fixed training set size $P = 128$, training and test loss diverge at a timescale ($\tau_{\mathrm{mem}}$) depending on $\sigma$ (inset), which scales as $\sigma^{-1}$ (main). *Right*: at fixed $\sigma^2 = 3.2 \cdot 10^{-2}$, $\tau_{\mathrm{mem}}$ increases with $P$ (inset), consistently with the scaling $\tau_{\mathrm{mem}} \propto P$ (main).

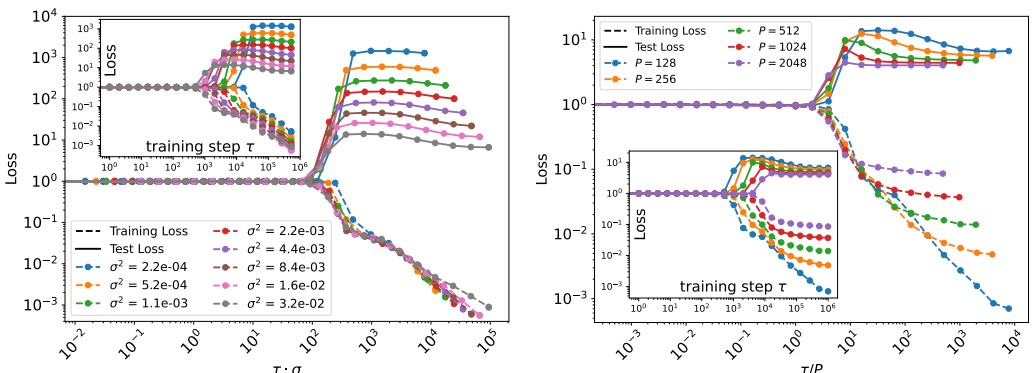

Figure 17: **Feature learning (mean-field) initialization, same setting as Figure 16.** Also in this case, $\tau_{\mathrm{mem}}$ is compatible with the scaling $\tau_{\mathrm{mem}} \sim \sigma^{-1}$ at fixed $P$ (*left*), and $\tau_{\mathrm{mem}} \propto P$ at fixed $\sigma$ (*right*).

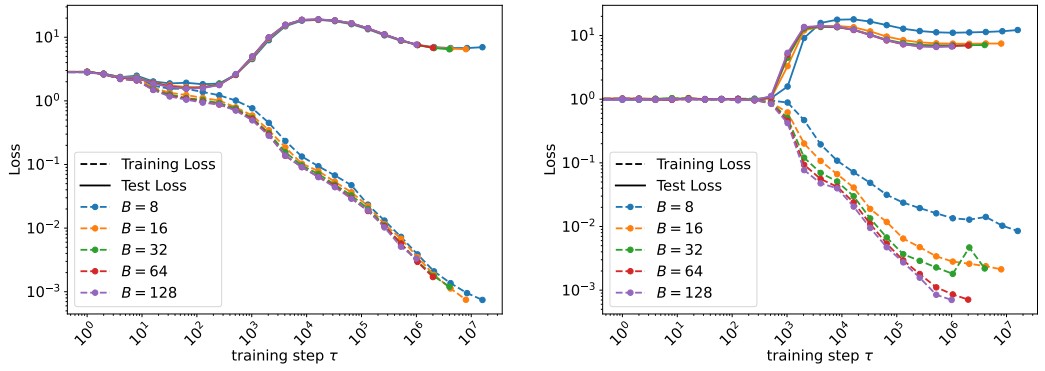

Figure 18: **Effect of changing batch size $B$, same setting as Figs. 16 and 17** (fixed $\sigma^2 = 3.2 \cdot 10^{-2}$, $P = 128$). Varying the batch size $B$ of training, both with the NTK (*left*) and feature learning (*right*) initialization, does not affect $\tau_{\mathrm{mem}}$.

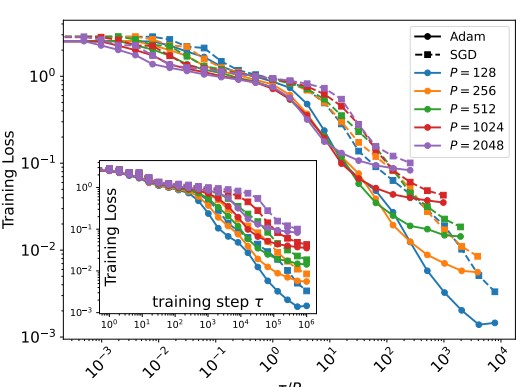

Figure 19: **Optimizer dependence.** Training loss as a function of training step for a one-hidden-layer ReLU neural network (NTK initialization) trained with Adam and SGD at multiple dataset sizes $P$. For each optimizer, we choose its maximal stable learning rate, defined as the largest learning rate for which the training loss reliably converges. For both optimizers, the curves at different $P$ start decaying at the same point when plotted against $\tau/P$, indicating the same linear scaling of the characteristic training time with $P$. Adam and SGD differ only by a horizontal shift, corresponding to an $\mathcal{O}_P(1)$ change in the constant of proportionality. The overall scaling and decay of the loss remain essentially identical. The inset shows the same data as a function of the unscaled training step $\tau$.

