# OpenReview forum: "Bigger Isn’t Always Memorizing: Early Stopping Overparameterized Diffusion Models"
_ICLR.cc/2026/Conference — Submitted to ICLR 2026_

### Official Review · Reviewer_hxnA · 2025-10-30

**Soundness:** 2
**Presentation:** 2
**Contribution:** 2
**Rating:** 4
**Confidence:** 4

**Summary:**

This paper analyze the learning dynamics of highly overparameterized diffusion models, which are supposed to memorize training data when sufficiently optimized. However, during training they first **generalize before memorizing/overfitting**. Empirically, the authors investigate image and language diffusion models on small datasets and consistently observe this early-stopping generalization. In this regime, the model achieves imperfect generalization with relatively lower validation loss, novel but lossy generations, and partial reproducibility. Motivated by these observations, the authors propose an early-stopping metric $\tau_{\text{mem}} \propto P$.

Finally, the authors provide a Random Hierarchy Model (RHM) perspective on this early-stopping generalization: according to previous work, to learn the $\ell$-th layer in the RHM model, one needs exponential data size $m^{\ell+1}$. With limited samples, the model only learns lower-level structures, which prevents full generalization.

**Strengths:**

1. The authors investigate the learning dynamics [1] of diffusion models on both images and language, providing insights into generalization vs. memorization and the training process of diffusion models. They also run a broad set of experiments supporting their arguments.
2. With the RHM model, the authors aim to characterize learning dynamics as learning different levels of dataset structure, i.e., **how much data is required to learn a given structural level**, aligning with coarse-to-fine learning behavior [1].

[1] Wang, Binxu. *An analytical theory of power law spectral bias in the learning dynamics of diffusion models.* NeurIPS 2025.

**Weaknesses:**

1. **A comprehensive and practical ablation on $\tau_{\text{mem}}$ is missing.** For instance, I would expect a clear scaling/regression plot for $\tau_{\text{mem}}$–$P$ validating the linear relationship, and analysis of how the coefficients depend on the data distribution and model size.

2. **The RHM theory does not fully justify memorization.** It explains partial generalization before memorization, but several definitions are missing: What is an empirical version of RHM data? How are errors at different levels $\ell$ in Figure 5 computed, and are they train or test errors? There is no rigorous distinction between empirical and population losses within RHM.

   As a result, The kernel-regression setup (Sec. 3.3) introduced to explain memorization and the linear dependency $\tau_{\text{mem}} \propto P$ feels disconnected from the RHM story.

3. **Some claims are not fully supported by experiments.** “At some time $\tau_{\text{mem}}$, the models begin to diverge. This divergence coincides with the onset of memorization” (L200–202). In Figure 2, inter-model similarity decreases monotonically, while similarity to training data increases monotonically; $\tau_{\text{mem}}$ is not clearly special. “With direct implications for hyperparameter transfer and privacy-sensitive applications” (L26–27). I do not see a straightforward justification—please elaborate (e.g., via Stable Diffusion experiments or deeper analysis as in point 1).

**Questions:**

1. In the caption of Figure 1, you state “$\tau_{\text{mem}}$ scales approximately linearly with $P$.” Is this driven by using equal intervals for $P=\{2048,4096,8192,16384\}$?
2. How does $\tau_{\text{mem}}$ change with different optimizers and schedules (e.g., AdamW vs. Adam, warmup/cosine)? This seems crucial for practicality.
3. For language diffusion models, can you provide some generated text samples (pre- and post-$\tau_{\text{mem}}$) for illustration?

*My current rating is a provisional assessment and may be updated after author responses and discussion with other reviewers.*

---

> ### Author Response · Authors · 2025-11-24
>
> We thank the reviewer for their assessment of our work. Below, we address each of the raised points in detail.
>
> **1.1 + Q1. Scaling $\tau_{\rm mem}$**
>
> We would like to remark that the linear scaling of $\tau_{\rm mem}$ with the dataset size $P$ is already shown **quantitatively** in the insets of Figures 1, 3, and 4. This effect is *not* an artifact of using "equal intervals" of $P$: (i) the values of $P$ we use are not equally spaced, and (ii) the key evidence comes from the fact that, when we plot training curves against $\tau / P$, all curves collapse. This standard rescaling procedure is a quantitative test of the relation $\tau_{\rm mem} \propto P$. Moreover, this scaling is supported by our kernel-based analysis, which yields $\tau_{\rm mem} \propto P$ without any assumptions about the data distribution.
>
> In response to the reviewer’s request, we have now added plots of $\tau_{\rm mem}$ vs. $P$ for images and language as well in the revision (Figure 9), with linear fits. They reinforce the linear scaling between the two quantities.
>
> **1.2 Dependence on model size and data distribution**
>
> Our focus is on the overparameterized regime. When models are initialized and scaled appropriately (e.g., via $\mu$-P), their behavior converges to a well-defined infinite-width limit, and our conclusions about the scaling of $\tau_{\rm mem}$ become effectively independent of model size.
>
> Regarding data distribution, in realistic applications, one typically cannot vary or control the underlying distribution: natural datasets (i) come as given and (ii) their internal structure is unknown. Our goal is hence studying how $\tau_{\rm mem}$ scales with $P$ across different datasets and modalities. Empirically, we observe the same linear scaling law across images, language, RHM, and Gaussian data, which suggests a robust, universal feature of the dynamics.
>
> Motivated by the reviewer's question, we computed the dependence of $\tau_{\rm mem}$ on data dimension $d$ in our theoretical arguments for kernel methods. We found that $\tau_{\rm mem} \sim C(d)^{-1} d^{1 - \nu/2} \ P/\sigma^\nu$ (as $d \to +\infty$), where $C(d)$ is a coefficient depending on the kernel and the data dimension. The full computation is reported in the updated version of the Appendix. We point out that this result is distribution-agnostic as it simply uses the fact that training points are isolated. Thus, as long as diffusion happens in the ambient space, the same argument applies to data supported on a lower-dimensional manifold, so the intrinsic dimension of the data does not affect our results, which only depend on $d$.

---

> > ### Author Response · Authors · 2025-11-24
> >
> > **2. Random Hierarchy Model analysis**
> >
> > The Random Hierarchy Model is a probabilistic context-free grammar (PCFG). Given randomly sampled production rules, the RHM defines a language (and probability distribution) over sequences of length $d$, with an exponentially large set of valid strings.
> >
> > - **Empirical RHM data.** In our experiments, we draw a finite, small set of sentences from this grammar and use them as the training set. The empirical distribution is the uniform distribution over these sampled strings, exactly as for natural data.
> >
> > - **Population vs. empirical loss.** The population distribution is the full distribution induced by the grammar rules. An estimated population loss is computed on held-out samples from the same grammar. The empirical loss instead is computed on the finite training set. Both are shown in the top panel of Figure 4 and display different behaviors during training, illustrating the distinction between empirical and population losses in this setting.
> >
> > - **Layer-wise errors.** The level-$\ell$ errors are defined by checking whether sequences generated by the diffusion model satisfy the constraints induced by the RHM grammatical rules at level $\ell$. These quantities are evaluated on generated samples, and thus measure how much of the underlying hierarchical structure is captured in the model’s output at each time. We will make these definitions explicit in the revised manuscript.
> >
> >
> > The RHM provides a controlled setting in which we can quantitatively measure which levels of structure (grammar layers) are learned before memorization.
> >
> > Regarding the kernel-regression setup, our goal is to analyse a regime that is more general than the RHM and not restricted to a particular data structure, because the empirical scaling $\tau_{\rm mem} \propto P$ is observed consistently across all datasets we consider -- natural images, language, RHM, and Gaussian data. In this sense, the kernel analysis is a distribution-agnostic theory that explains this scaling for score fitting. The RHM serves a complementary purpose: it gives a structured synthetic environment where we can interpret the statistics of early-stopped generations rather than just their loss or similarity statistics. We will clarify in the paper that we use: (i) kernel regression to explain the universal scaling of $\tau_{\rm mem}$; (ii) the RHM to interpret partial generalization and early stopping in terms of data statistics.
> >
> >
> > **3.1 Celeb-A experiment**
> >
> > In the CelebA experiment (Figure 2), we define $\tau_{\rm mem}$ as the time at which the training and validation losses first bifurcate, consistent with the rest of the paper. We thank the reviewer for pointing out that this was not fully explicit and have clarified it in the revision.
> >
> > Concerning the curves in Figure 2: the inter-model similarity and the similarity to the nearest training image are both monotonic, but their relative behavior identifies a meaningful transition. Up to $\tau_{\rm mem}$, the cosine similarity between the outputs of the two models (blue) is larger than the similarity to any training image. This indicates that the two models, trained on disjoint datasets, are producing similar images to each other, rather than copies of their own training data. After $\tau_{\rm mem}$, the blue curve drops below the similarity to training images, and the similarity to the respective training sets continues to increase. This marks the onset of memorization. We will highlight this crossing behavior more clearly in the text, as it is the sense in which $\tau_{\rm mem}$ is a special time in Figure 2.
> >
> > **3.2 Practical implications**
> >
> > Our results suggest two concrete applications.
> >
> > - **Hyperparameter transfer.** The empirical law $\tau_{\rm mem} \propto P$ gives a quantitative rule for scaling training time with dataset size to avoid entering the memorization regime. In practice, this means that once $\tau_{\rm mem}$ has been identified for a given model/dataset size, one can extrapolate the early-stopping time for larger or smaller datasets without redoing extensive trial-and-error searches. We used this principle ourselves when planning our experiments (e.g., deciding how long to train for different $P$).
> >
> > - **Privacy-sensitive applications.** Our analysis shows that early stopping at $\tau_{\rm mem}$ significantly reduces copying, both at the pixel level and in the SSDC-based feature space for Stable Diffusion, where the similarity between generated images and training images approaches that of validation images. This directly impacts scenarios where copying may cause copyright or privacy issues (e.g., training on proprietary or sensitive data): the model can be trained in a regime where memorization is empirically controlled by an early-stopping time (that scales predictably with $P$).

---

> > > ### Author Response · Authors · 2025-11-24
> > >
> > > **Q2. Dependence on optimizers and schedules**
> > >
> > > Empirically, we found the linear $\tau_{\rm mem} \propto P$ relation to be highly robust to optimization choices:
> > >
> > > - In our image, language, and RHM experiments, we use Adam and observe the same scaling.
> > > - For RHM, we also ran experiments with SGD and observed the same phenomenology.
> > > - In Section F, when validating our theoretical scaling with kernel regression, we use both full-batch gradient descent and vanilla SGD with different batch sizes; in all cases, $\tau_{\rm mem}$ scales linearly with $P$.
> > > - In the Stable Diffusion experiments, we use AdamW with a warm-up schedule, and the scaling law persists.
> > >
> > > We added a comment in the submission.
> > >
> > > **Examples for language diffusion models** Following the reviewer’s suggestion, we have added examples of text generated by the language diffusion model in the Appendix.

---

> ### Comment · Reviewer_hxnA · 2025-11-26
>
> I thank the authors for their effort in the rebuttal, specifically for providing the text diffusion examples and clarifying $\tau_{mem}$ as a crossover time in similarity. However, I still retain a few concerns:
>
> 1. The layer-$\ell$ loss still primarily reflects imperfect generalization. Since both a fully memorized model and a generalized model can achieve small layer-$\ell$ loss for some $\ell$, this metric does not intuitively tell you if a model is reproducing training examples.
> 2. Justification for $\tau_{mem} \propto P$: regarding the model-agnostic justification, a simpler intuition seems sufficient. If we assume the gradient step a single sample receives is proportional to $1/P$, the overall steps needed to learn local attraction to that sample would naturally be proportional to $P$. (This appears to align with the proof idea in the kernel justification).
> 3. On the robustness to optimizer, my previous concern is that for **the same training set**, the coefficient can change drastically when applying different optimizers (Adam v.s. SGD). This can add significant complexity to this proportional behavior. I also agree with the other reviewers that the impact of different underlying data distributions is a valid concern.
>
> In light of the clarifications in the rebuttal, I can raise my score to 6 but lower my confidence if the authors would like.

---

> ### Author Response · Authors · 2025-11-27
>
> We thank the reviewer for their reply.
>
> 1\. We agree that the **layer-$\ell$ error** by itself is not a direct
> measure of generalization. We do not make such a claim in the paper.
> Instead (cf., e.g., L403), in the RHM setting, generalization
> corresponds to (i) low layer-wise errors *together with* (ii) a
> vanishing fraction of copies. In Figure 5-(a), we report an experimental
> demonstration of this, as noted in the caption: "However, the rules at
> the deepest level $L = 5$ are never learned, and the corresponding error
> decreases only when memorization occurs, since $P = 1024$ is smaller
> than the sample complexity $P^*_L \sim 10^4$". The role of the
> layer-wise errors is thus to quantify, during training, whether generated data is consistent with the hierarchical rules, not to
> diagnose generalization on their own. In addition, we always report the
> **validation loss**, which is an estimate of the population loss on
> held-out data. For diffusion models, this loss is a lower bound on the
> **log likelihood** and thus a **proper measure of generalization** that
> should address the reviewer's concern.
>
> 2\. The reviewer's intuition is consistent with the observed scaling,
> but implicitly puts together several nontrivial assumptions: (i) that
> each training point is memorized independently of all others, (ii) noisy
> observations of a training point are effectively treated as the same
> input, and (iii) memorizing a point requires $\mathcal{O}(1)$ gradient
> steps, independently of the global evolution of the loss. Our kernel
> regression analysis makes **explicit assumptions** and
> **quantitatively** relates the scaling of $\tau_{\rm mem}$ to the
> spectrum of the empirical kernel. In particular, $\tau_{\rm mem}$ is
> controlled by the relevant eigenvalues of the kernel (those associated
> with fitting localized linear modes). We show its dependence not only on
> $P$ but also on the kernel shape and input dimension. So while the
> reviewer's intuition gives, a posteriori, the correct scaling, our
> scaling theory explains **when and why this scaling should be expected
> to hold and why it is robust**, e.g., to the data distribution, in the
> experiments we report.
>
> 3\. Our main claim is about the **scaling of the memorization time** with the training set size,
> not the exact coefficient: if the memorization time satisfies
> $\tau_{\rm mem} \propto P$ while the onset of generalization occurs at
> $\mathcal{O}(1)$ training time, then a window of
> "generalize-then-memorize" dynamics necessarily exists, regardless of
> the proportionality constant. In practice, across all settings we have
> studied -- using the **standard hyperparameters** from the original
> works (**OpenAI**'s settings for iDDPM, **DeepMind**'s for MD4,
> **StabilityAI**'s for Stable Diffusion) -- we find that the
> proportionality coefficient is in a relatively narrow range roughly
> between 1 and 10, so there is no evidence of "significant added
> complexity". For these SOTA settings, changing optimizer (e.g., from
> AdamW to SGD) can result in convergence issues and is not practically
> viable. For simpler architectures, such as the UNet used with the RHM or
> the shallow networks in our kernel experiments, we did compare **Adam**
> vs. **SGD** (using the maximally stable learning rate for each setting)
> and observed either the same (RHM, Fig. 15) or small differences (NTK,
> Fig. 19) in the prefactor. These results confirm the robustness of our
> findings to the optimizer choice.
>
> Regarding the **data distribution**: In the paper, we spanned a broad
> set of distributions. Our experiments cover **CIFAR**, **LAION**,
> **text8**, a **probabilistic context-free grammar** with random
> production rules (RHM), **Gaussian mixtures**, and partial
> generalization on **CelebA**. This diversity is, to our knowledge,
> unusually broad for a single study of diffusion dynamics, and in all
> these cases we observe the same qualitative picture and the same linear
> $\tau_{\rm mem} \propto P$ law. From a theoretical perspective, our
> **kernel analysis is distribution-agnostic** under mild assumptions
> (separation of points and diffusion in ambient space) and does not
> depend on specific properties such as intrinsic dimension. Given that in
> real applications the underlying data distribution is fixed and not
> directly controllable, we see this combination of broad empirical
> coverage and distribution-agnostic theory as a very strong answer to the
> reviewer's concern.
>
> We hope our answers have further addressed the reviewer's new concerns,
> and kindly ask them to reconsider their score while we remain available
> for further discussions.

---

### Official Review · Reviewer_zPRo · 2025-11-01

**Soundness:** 4
**Presentation:** 4
**Contribution:** 2
**Rating:** 4
**Confidence:** 4

**Summary:**

The paper investigates training dynamics of diffusion models with respect to generalization and memorization. Empirically authors show that diffusion models first learn to generate samples from *entire* data distribution (generalization), and after certain point it learns to generate samples from *training* data distribution (memorization).

**Strengths:**

- The presentation is clear.
- The fact that diffusion models first try to generalize before memorizing is a new observation.
- The authors conducted extensive experiments on various modalities.

**Weaknesses:**

- As in [Deep Double Descent](https://arxiv.org/abs/1912.02292), the number of training epochs is also included in training capacity. Hence the fact that memorization time and dataset size having linear dependency is not very surprising.
- The observation might not be very practically applicable, because most practical vision diffusion models are trained on very large dataset. Also since it's been reported that memorization happens at the *concept* level, it would be very hard to quantify *validation error* and hence the correct *early stopping point*.
- The paper is primarily scientific report, where most contents align with the existing perspective.

**Questions:**

- Is the FID score at $\tau_\text{mem}$ comparable to FID score of same model trained on whole dataset?
- If so then would this imply that even 2048 images are enough to represent the whole cifar10 dataset?

---

> ### Author Response · Authors · 2025-11-24
>
> We thank the reviewer for their comments. Below, we address them point by point.
>
> **Relation to supervised learning and double descent**
>
> We respectfully but strongly disagree with the fact that our findings are not surprising. In particular, they are not a straightforward consequence of existing supervised-learning intuitions. In supervised learning, one can have *benign overfitting*: overparameterized models perfectly fit the training labels and still generalize well to unseen data. This is the regime of the second descent in double descent, also mentioned by the reviewer. In contrast, for overparameterized diffusion models, minimizing the training loss corresponds to perfectly fitting the empirical score function, which in turn implies memorization of the training points and precludes any generalization. In other words, benign overfitting is impossible. We refer to the paragraph "Overfitting in supervised learning vs. diffusion models" in the submission for a longer discussion on this matter.
>
> Moreover, even the seemingly simple observation that $\tau_{\rm mem}$ scales linearly with $P$ hides nontrivial mechanisms. For instance, a naive explanation based on each data point being seen fewer times when $P$ grows would predict a strong dependence on batch size. Instead, as we show (L274), the memorization time measured in optimization steps is independent of batch size. This includes the extreme cases of fixed small batch size and full-batch Gradient Descent. In the former case, memorization happens when the model has seen each data point a fixed number of times; in the latter case, instead, each data point is seen a number of times proportional to $P$. In both cases, the memorization time, measured in terms of the number of training steps, is identical, indicating a more complex dynamics than simple arguments would suggest.
>
> **Practical implications**
>
> We respectfully disagree that our observations have limited practical relevance. First, our Stable Diffusion experiments directly target *fine-tuning* foundation models, where practitioners routinely work with relatively small, curated datasets (e.g., domain adaptation, model personalization, scientific applications). In this setting, understanding when memorization occurs and how it scales with $P$ is directly actionable, and our results show that early stopping can substantially reduce copying. Second, even in pretraining, the regime under which state-of-the-art models operate is less clear than suggested. Papers or technical reports rarely measure memorization. For instance, the EDM-2 models by Karras et al. (2024) at NVIDIA use architectures with $\sim 10^9$ parameters trained on ImageNet ($\sim 10^6$ images), which fall within the overparameterized regime we study. Yet, copying is not analyzed there. At the same time, several works - summarized in our introduction and related work section - have provided evidence of copying in large-scale vision diffusion models, indicating that memorization is not purely a small-data issue. We will strengthen this discussion to connect our findings to both pretraining and fine-tuning practice more clearly.
>
> **Concept-level memorization**
>
> Our analysis already goes beyond exact pixel-level copying in several ways. The validation loss we track is a lower bound on the log-likelihood and thus measures how well the model fits the full distribution, not just exact reproductions. Moreover, for Stable Diffusion (Figure 7), we use SSDC (Self-Supervised Descriptor for Image Copy Detection), a similarity measure in the latent space of trained neural networks. This metric is sensitive to semantic similarity, not only pixel-level matches, and has been used to detect near-duplicate and concept-level copies in challenging datasets such as LAION (e.g., Somepalli et al., 2023).
>
> Empirically, we find that early stopping at $\tau_{\rm mem}$ minimizes SSDC, bringing the similarity between generated and training images down to the same order as the similarity between validation images themselves. This shows that our early stopping criterion is already effective in reducing concept-level copying, not only exact memorization.
>
> We note that, interestingly, for the RHM we observe that rules across all hierarchy levels are memorized jointly when memorization starts. The fact that, in practice, concept-level memorization sometimes appears may be related to underparameterization, language conditioning, or crossover phases between generalization and memorization. Exploring these hypotheses is an exciting direction for future work.

---

> > ### Author Response · Authors · 2025-11-24
> >
> > **Alignment with existing perspectives**
> >
> > Our work is primarily scientific in nature, and we see this as a strength rather than a limitation. It provides a systematic, cross-modality study of memorization in overparameterized diffusion models, backed by a quantitative theory. As the reviewer themself notes in the strengths section, the empirical observation that diffusion models "first try to generalize before memorizing" is new, and our experiments on multiple modalities (images, discrete text) and synthetic grammars go beyond existing perspectives. In particular, we contribute:
> > (i) a unified picture of temporal phases of generalization vs. memorization for diffusion models;
> > (ii) a scaling theory explaining $\tau_{\rm mem} \propto P$ in terms of score-fitting dynamics; and
> > (iii) a synthetic framework (RHM) that makes partial generalization and the cost of early stopping quantifiable.
> > Additionally, as argued above, our results are practically relevant, encompassing both pre-training and fine-tuning of foundation diffusion models and concept-level copying.
> >
> > **FID and image quality**
> >
> > Regarding the question on FID at $\tau_{\rm mem}$: on CIFAR-10 with P = 16k training images, we report an FID of $\approx 5$ at $\tau_{\rm mem}$. The original iDDPM paper reports an FID of $\approx 3$ when training the same architecture on the full training set with tuned hyperparameters. We therefore obtain a competitive but not identical FID at early stopping with fewer data and without aggressively optimizing hyperparameters. Importantly, as shown in Figure 9, in our setup, FID reaches its minimum around $\tau_{\rm mem}$, indicating that early stopping jointly minimizes FID and memorization.
> >
> > Concerning whether this would imply that 2k images "are enough to represent the whole CIFAR-10 dataset": the answer is no. Models trained on 2k images have noticeably higher FID, and their samples at the early stopping time are in the partial/progressive generalization regime, qualitatively similar to those illustrated in Figure 2. In this case, early stopping still effectively prevents copying, but the resulting model does not match the performance of one trained on the full dataset. We will clarify these points and explicitly distinguish between the two regimes in the revision.

---

### Official Review · Reviewer_bGKP · 2025-11-02

**Soundness:** 3
**Presentation:** 3
**Contribution:** 2
**Rating:** 4
**Confidence:** 3

**Summary:**

This paper investigates the dynamic transition from generalization to memorization of diffusion models during the training process. The authors demonstrate across image and language models that generalization is, in fact, achieved progressively during training before the onset of memorization, finding an empirical law that the memorization time is proportional to the dataset size. Ultimately, the results suggest that generalization and memorization are distinct temporal phases, implying that a principled, dataset-size-aware early-stopping criterion can be an optimal strategy for preserving generalization and avoiding memorization in large diffusion models

**Strengths:**

Strengths:
* Besides image generation, they also study the memorization of the masked diffusion model in the text modality, which is novel in the generalization-memorization field to me.
* They use the reproducibility of two different models trained over two disjoint datasets to show that the score function attempted to learn the real underlying distribution at the early stage.
* The random hierarchy model further provides some interesting insights to the learning process, such as the partial generation.

**Weaknesses:**

Weaknesses:
* In Section 3.1, you showed that the transition point $\tau_{mem}$ scales approximately linearly with the training set size. Do you think the distribution complexity, such as the intrinsic dimension of the data and the entropy of the data, also influences the transition point? Besides, for the latent diffusion model and pixel diffusion model, is there any differences on the transition point? Including more factors into your study would make this work more thorough and robust.
* The key claim of this paper is that the model first generalizes at an early stage but then memorizes after $\tau_{mem}$. But I am concerned about calling the first stage generalization. Although the validation loss indeed decreased in the first stage, it may still be too high, and the score function hasn’t learned a good distribution. As you also visualized in Figure 2 (right), the generated images before $\tau_{mem}$ have bad quality and thus cannot be treated as good generalizations. Then, the early stopping strategy fails in this case. I feel that the early stop is effective only when both the sample size and network size are large, which is also supported by Figure 6.
* The dynamic transition and the linear relation between $\tau_{mem}$ and dataset size have also been revealed in prior work [1]. Could you compare the novelty of your work?

[1]: Why Diffusion Models Don't Memorize: The Role of Implicit Dynamical Regularization in Training. https://arxiv.org/abs/2505.17638

**Questions:**

Do you also visualize the partial generalization in real images? Is there any prior work proposing hierarchy data frameworks for images?

---

> ### Author Response · Authors · 2025-11-24
>
> We thank the reviewer for their constructive feedback. Below, we address the questions and concerns that were raised.
>
> **Scaling of $\tau_{\rm mem}$ with data properties**
>
> We agree that, in principle, $\tau_{\rm mem}$ could depend on properties of the data distribution. In practice, however, data is given, and the main controllable variable is the training set size, which our analysis focuses on across multiple data types and settings.
>
>  Nonetheless, motivated by the reviewer's question, we computed the dependence of $\tau_{\rm mem}$ on data dimension $d$ in our theoretical arguments for kernel methods. We found that $\tau_{\rm mem} \sim C(d)^{-1} d^{1 - \nu/2} \ P/\sigma^\nu$ (as $d \to +\infty$), where $C(d)$ is a coefficient depending on the kernel and the data dimension. The full computation is reported in the updated version of the Appendix. We point out that this result is distribution-agnostic as it simply uses the fact that training points are isolated. Thus, as long as diffusion happens in the ambient space, the same argument applies to data supported on a lower-dimensional manifold, so the intrinsic dimension of the data does not affect our results, which only depend on $d$.
>
> **Latent diffusion models**
>
> To address the question about latent vs. pixel diffusion, we have repeated the Stable Diffusion fine-tuning experiment of Appendix C while varying the number of training images (Figure 7 of the updated manuscript). We again observe the linear scaling, supporting the universality of the phenomenon across both pixel and latent diffusion models and further strengthening our claims.
>
> **Progressive generalization**
>
> By generalization, we refer to the model's score approaching the population score, as measured by decreasing validation loss and agreement between models trained on disjoint datasets, rather than to the perceptual quality of samples alone. We agree that, for small $P$, the validation loss at early times may still be high and the generated images in Figure 2 appear low-quality. This is, in fact, the point of that experiment. We are not claiming the generalization is "good" or "complete". Rather, we show that the two models produce nearly identical, low-quality images. This is strong evidence that they are both learning the same score function – i.e., generalizing – rather than memorizing their respective datasets. This is why, as we wrote in L277–279, Section 4 introduces synthetic grammars to quantify the accuracy of generations under limited training. Indeed, in the RHM setting, we can explicitly track which grammar rules have been acquired before $\tau_{\rm mem}$, thus making partial generalization precise. Early stopping recovers only a subset of the hierarchical structure (lowest levels) when data is scarce, and more as $P$ grows. Therefore, it is always effective for preventing memorization, but only when the dataset is sufficiently large, the early-stopped model also achieves high-quality generations – as summarized in the phase diagram of Figure 6. We will clarify this distinction in the main text.
>
> **Concurrent work**
>
> The work suggested by the reviewer is contemporaneous with our submission (both appeared online at the same time) and, nevertheless, is already cited and contrasted in our paper. Notice that, per ICLR guidelines, contemporaneous work should not be used as a weakness or taken into account in the evaluation. Moreover, our work provides several distinct and significant contributions beyond [1]: (i) we conduct larger-scale experiments on image diffusion models, including Stable Diffusion, and additionally study memorization in language diffusion models, which to our knowledge had not been empirically characterized before; (ii) our scaling theory covers any isotropic kernel and is not restricted to random-features; (iii) we use the RHM to formally define and quantify partial generalization and the cost of early stopping, which directly explains the low-quality generations the reviewer observed. Finally, we connect these insights in the phase diagram of Figure 6.
>
> **Partial generalization in real images**
>
> Figure 2 is the visualization of this phenomenon in real images. The low-quality, inconsistent images shown in Figure 2 (right panel, $\tau < \tau_{\rm mem}$) demonstrate this concept. The models have clearly learned some shared, underlying structure, like low-level features. However, they have not yet learned the full set of "rules" to make a face coherent, i.e., the global structure. This is a direct visual analog to the partial generalization we formalize with the RHM, where a model might learn the rules for the first layers, but not the deeper, global rules.

---

> > ### Author Response · Authors · 2025-11-24
> >
> > **Hierarchical frameworks for images**
> >
> > There is indeed a long line of work modeling images and scenes via discrete, compositional hierarchies, often inspired by parsing trees in theoretical linguistics, e.g., pattern theory (Stoyan, 1997; Jin \& Geman, 2006; Siskind et al., 2007; Li et al., 2009). These models hierarchically decompose visual scenes into objects, parts, and primitives. The RHM can be viewed as an abstracted, controllable version of such grammars for images. We will add these references and clarify this connection in the submission.

---

### Author Response · Authors · 2025-11-30
**Summary for the AC**

We thank the reviewers for their time and for providing feedback on our work. We appreciate that reviewers highlighted the clarity (zPRo), novelty (zPRo), breadth of experiments across modalities (zPRo, hxnA) - including the first study of memorization in diffusion LLMs (bGKP) -, and the insights on generalization via probabilistic context-free grammars (bGKP, hxnA).

We summarize the main clarifications from the rebuttal below.

- **Scaling of memorization time with data properties**

In the paper, we quantify - both **experimentally** on SOTA systems and **theoretically** by studying kernel methods - the scaling of the **memorization time** $\tau_{\rm mem}$ in diffusion models with the training set size $P$. Some reviewers were concerned with the impact of data distribution on our results. We clarified that the linear scaling $\tau_{\rm mem} \propto P$ holds empirically across **diverse datasets** (CIFAR, LAION, text8, RHM, Gaussian mixtures), and that our kernel-based theory (numerically validated also for neural networks beyond the Neural Tangent Kernel regime) explains this scaling in a **distribution-agnostic** way, depending only on ambient (and *not* intrinsic) dimension under mild assumptions. This strongly supports the claim that, in realistic settings where the data distribution is given, $\tau_{\rm mem}$ is controlled by $P$.

- **Relation to supervised learning, double descent, and existing perspectives**

We emphasized that, unlike supervised learning with benign overfitting, minimizing the diffusion training loss implies perfect fitting of the empirical score and hence memorization: Double descent and benign overfitting are impossible. Our work provides a unified **"generalize-then-memorize"** picture for diffusion models, quantifies $\tau_{\rm mem} \propto P$ via experiments and kernel theory, and uses the **RHM to formalize partial generalization** and the cost of early stopping. All these are novel contributions that challenge prevailing ideas that diffusion models generalize only when they fail to memorize, e.g., due to finite capacity or architectural constraints. Concerning the differences with [1], NeurIPS 25 best paper award, which is *contemporary* to our work and finds similar conclusions, we refer to the answer "Concurrent work" to Reviewer bGKP.

*[1] Why Diffusion Models Don't Memorize: The Role of Implicit Dynamical Regularization in Training. arxiv:2505.17638.*

- **Practical implications**

We argued that our findings are directly **actionable for fine-tuning** foundation models (e.g., Stable Diffusion) on small curated datasets, where early stopping at $\tau_{\rm mem}$ substantially reduces pixel- and concept-level copying (via SSDC), and that $\tau_{\rm mem} \propto P$ enables **hyperparameter transfer** of training time across dataset sizes. We also noted that many large-scale diffusion models still operate in **overparameterized** regimes where memorization is relevant but often unmeasured.

- **Robustness to optimizers and schedulers**

We showed that the linear $\tau_{\rm mem} \propto P$ law is **robust across optimizers** (Adam, AdamW, SGD), batch sizes (including extreme cases of small batch size and full-batch GD), and learning-rate schedules (with and without warm-up), indicating that the observed scaling reflects a **universal property** of diffusion training dynamics rather than a peculiarity of specific hyperparameters.

---

### Meta-Review · Area_Chair_K6xT · 2026-01-06

**Summary:**

The key decision factors were (i) whether the generalize–memorize dynamics in overparameterized diffusion models are convincingly established across settings, (ii) how robust and interpretable the proposed scaling law for memorization time is, and (iii) the practical meaning and actionability of early stopping, particularly with respect to partial generalization and concept-level memorization. Reviewers broadly agreed that the paper presents a careful and unusually broad empirical study across modalities (images, text, and synthetic grammars) and that the observed “generalize-then-memorize” training dynamics are real and scientifically interesting. However, the final assessment hinged on questions about how novel these findings are relative to contemporaneous work, how stable and interpretable the proportionality constant in the scaling law is, and whether early stopping constitutes a sufficiently concrete and scalable intervention for real-world diffusion training.

**Reviewer Concerns:**

Addressed concerns:
The rebuttal substantially strengthened the empirical and conceptual grounding of the work. The authors added explicit linear fits for the memorization time scaling, extended experiments to latent diffusion models, and demonstrated robustness across optimizers, batch sizes, and learning-rate schedules, supporting the claim of universality. They clarified the definition of “generalization” in diffusion models in terms of population/validation loss and reproducibility rather than perceptual quality alone, and formalized partial generalization using the Random Hierarchy Model. The rebuttal also clarified the distinction between partial generalization and sample quality, added text diffusion examples, and expanded discussion on concept-level copying using SSDC in Stable Diffusion, addressing several technical and interpretational concerns raised by reviewers.

Remaining concerns:
Despite these clarifications, some reviewers continued to view the contribution as primarily scientific rather than practically actionable. In particular, there remains skepticism about how useful early stopping is at large scale, given that many production diffusion models are trained on massive datasets and that identifying the correct stopping point in practice may be difficult. Reviewers also questioned whether the variability of the proportionality constant in the linear scaling law limits its usefulness for hyperparameter transfer, and whether all forms of concept-level memorization are fully captured by the proposed diagnostics. Finally, some reviewers perceived overlap with contemporaneous work on memorization dynamics in diffusion models and remained unconvinced that the incremental practical insight over that body of work is sufficient for acceptance.

**Reviewer Scores:**

bGKP: ~4

zPRo: ~4

hxnA: ~6

---

### Decision · Program_Chairs · 2026-01-26

Reject